



# Seasonal forecasting of lake water quality and algal bloom risk using a continuous Gaussian Bayesian network

Leah A. Jackson-Blake[1], François Clayer[1], Sigrid Haande[1], James Sample[1], Jannicke Moe[1]

[1]Norwegian Institute for Water Research (NIVA), 0349 Oslo, Norway

*Correspondence to*: Leah A. Jackson-Blake (leah.jackson-blake@niva.no)

## Abstract

Freshwater management is challenging, and advance warning that poor water quality was likely, a season ahead, could allow for preventative measures to be put in place. To this end, we developed a Bayesian network (BN) for seasonal lake water quality prediction. BNs have become popular in recent years, but the vast majority are discrete. Here we developed a

Gaussian Bayesian network (GBN), a simple class of continuous BN. The aim was to forecast, in spring, total phosphorus (TP), chlorophyll-a (chl-a), cyanobacteria biovolume and water colour for the coming growing season (May-October) in lake Vansjø in southeast Norway. To develop the model, we first identified controls on inter-annual variability in water quality using correlations, scatterplots, regression tree based feature importance analysis and process knowledge. Key predictors identified were lake conditions the previous summer, a TP control on algal variables, a colour-cyanobacteria relationship,

and weaker relationships between precipitation and colour and between wind and chl-a. These variables were then included in the GBN and conditional probability densities were fitted using observations (≤ 39 years). GBN predictions had $R^2$ values of 0.37 (cyanobacteria) to 0.75 (colour) and classification errors of 32% (TP) to 13% (cyanobacteria). For all but lake colour, including weather nodes did not improve predictive performance (assessed through cross validation). Overall, we found the GBN approach to be well-suited to seasonal water quality forecasting. It was straightforward to produce probabilistic

predictions, including the probability of exceeding management-relevant thresholds. The GBN could be purely parameterised using observed data, despite the small dataset. This wasn't possible using a discrete BN, highlighting a particular advantage of using GBNs when sample sizes are small. Although low interannual variability and high temporal autocorrelation in the study lake meant the GBN performed similarly to a seasonal naïve forecast, we believe the forecasting approach presented could be useful in areas with higher sensitivity to catchment nutrient delivery and seasonal climate, and

for forecasting at shorter time scales (e.g. daily to monthly). Despite the parametric constraints of GBNs, their simplicity, together with the relative accessibility of BN software with GBN handling, means they are a good first choice for BN development, particularly when datasets for model training are small.



## 1. Introduction

Despite their importance, freshwaters are under intense pressure from human activities. Severe declines in the quantity and
quality of habitats and species abundance are widespread, and freshwaters are now one of the most threatened ecosystem
types in large parts of the world (Dudgeon et al., 2006; Gozlan et al., 2019; Reid et al., 2019). To try to safeguard freshwater
condition, the EU Water Framework Directive (WFD) requires all waterbodies to achieve at least "Good" ecological status
by 2027, assessed using a set of indicators of ecosystem integrity (EC, 2003). However, meeting environmental targets is
challenging, and despite widespread implementation of measures to improve water quality, 60% of European surface waters
were still below "Good" ecological status in 2018 (Kristensen et al., 2018). Harmful cyanobacterial blooms are a particular
concern worldwide as they can produce harmful toxins, damage ecosystems, jeopardise drinking water supplies, fisheries
and recreational areas, and are becoming more widespread, frequent and intense due to eutrophication and climate change
(Huisman et al., 2018; Ibelings et al., 2016; Taranu et al., 2015).

Advance warning, a season in advance, that poor water quality was likely could allow for measures to be put in place to
reduce the impacts. For example, water levels could be raised or lowered in flow-regulated waterbodies, more stringent farm
management or effluent discharge advice could be issued, or measures could be taken to increase preparedness (for example
if problems with drinking water supply were expected). Although many cyanobacteria forecasting systems have been
developed, they all predict conditions at most a month in advance or focus on multi-decadal climate and land use change
impacts (reviewed in Rousso et al., 2020). Seasonal forecasts, issued with lead times of 1-6 months, could allow for more
comprehensive preventative or mitigative measures. Seasonal forecasting is a growing area of research, often taking
advantage of developments in seasonal climate forecasting, and there are many potential management applications (Bruno
Soares & Dessai, 2016). However, seasonal forecasting within the water sector has so far been largely limited to streamflow
forecasting, with only recent applications to lake water temperature (Mercado-Bettín et al., 2021) and none, to our
knowledge, to lake water quality.

To issue a seasonal forecast for summer (e.g. May-October) lake water quality, we need to first understand the key factors
controlling inter-annual variability in lake water quality. Here, we focus on three water quality indices used in WFD status
assessments in Norway: total phosphorus (TP), as P is usually the limiting nutrient for phytoplankton (although see e.g.
Dolman et al., 2012; Gobler et al., 2016); chl-a, as a basic indicator of algal biomass; and cyanobacterial biomass. We also
forecast lake colour, of relevance for drinking water treatment. Lake TP concentration and colour may be controlled by
delivery from the surrounding catchment, interaction with lake sediments, lake stratification and mixing (Søndergaard et al.,
2013; Welch & Cooke, 2005). Many studies have examined the drivers of algal biomass development in lakes and the causes
of harmful algal blooms. The right combination of environmental conditions, including sufficiently high nutrient
concentrations, in particular P (e.g. Heisler et al., 2008; Lürling et al., 2018; Stumpf et al., 2012), temperature (e.g. Kosten et



al., 2012; Paerl & Huisman, 2009; Robarts & Zohary, 1987), light intensity (e.g. Kosten et al., 2012; Merel et al., 2013), and a stable water column (e.g. Huber et al., 2012; Yang et al., 2016) can lead to cyanobacteria bloom formation. The relative importance of different drivers varies according to lake type, with nutrients often providing a dominant control in polymictic lakes, whilst dimictic lakes are generally more sensitive to climatic variables through their effect on water column stability

(Taranu et al., 2012). Because of the combination of factors that together control bloom formation, it is hard to make "one-size-fits-all" models, and models for predicting cyanobacteria bloom occurrence are therefore generally site specific (Rousso et al., 2020).

A multitude of potential methods exist for seasonal forecasting of water quality. Here, we adopt a Bayesian network (BN)

approach. BNs are a type of probabilistic multivariate model which is well suited to environmental modelling, risk assessment and forecasting (Aguilera et al., 2011; Kaikkonen et al., 2021; Uusitalo, 2007). In brief, BNs are graphical models in which the joint probability distribution among a set of variables $X = [X_1,…X_n]$ is represented in terms of: (1) a directed acyclic graph, where each vertex (or node) represents a variable in the model, and an edge (or arc) linking two variables indicates statistical dependence; (2) conditional distributions for each variable $X_i$, $p(X_i|p_a(X_i))$, given the

probability distribution $p_a(X_i)$ of any parent nodes, which quantify the strength and shape of dependencies between variables (Pearl, 1986). In recent years BNs have become popular in a broad range of environmental modelling disciplines, including modelling lake water quality and algal bloom risk (e.g. Couture et al., 2018; Gudimov et al., 2012; Rigosi et al., 2015; Shan et al., 2019; Williams & Cole, 2013). Particular strengths in terms of our seasonal forecasting aims are that, as nodes are modelled using probability distributions, risk and uncertainty can be estimated easily and accurately compared to many other

modelling approaches. They can thus be powerful tools to assess the probability of events (e.g. WFD ecological status class). They are also well suited for communicating and visualizing the model to end users and it is easy to update the model given new data. Other benefits include the ability to model complex systems in a quick and efficient way, to combine data and expert knowledge, easy handling of missing values, and the potential to be used for inference as well as prediction.

BNs were originally designed to deal with discrete data. Relationships between nodes in discrete BNs can be non-linear and complex, thereby allowing for the full power of BN analysis, and the vast majority of environmental BN models are discrete (Aguilera et al., 2011). Any continuous variables must first be discretized, but this involves an information loss as discretization can only capture the rough characteristics of the original distribution, and discretization choices (number of intervals and division points) affect BN results (e.g. Nojavan et al., 2017) and their interpretation (Qian & Miltner, 2015). In

practice it is usually necessary to restrict the number of intervals, often to just two or three classes, as the more intervals, the more data are needed to parameterise the model meaningfully (Hanea et al., 2015). However, such restrictions mean it becomes difficult to capture complex relationships, thereby diminishing the theoretical benefits of using a discrete network (Uusitalo, 2007). Continuous or hybrid BNs, where continuous nodes are allowed, avoid the need for discretization, and a number of new algorithms for non-parametric continuous networks have been developed in recent years (Marcot & Penman,



2019). However, Gaussian BNs (GBN) are a long-established, simple and powerful class of continuous BN, and are often the only type of continuous node available in commonly-used BN software (e.g. Bayes server, BNLearn, Hugin). In GBNs, each random variable is defined by a Gaussian distribution and variables are linearly related to their parents (Geiger & Heckerman, 1994; Shachter & Kenley, 1989). In some situations these parametric constraints may be overly-limiting, but when this approximation is appropriate GBNs may be preferable over discretization. Despite the potential benefits, the use of

continuous BNs in environmental modelling is rare. In a review of papers published over the period 1990-2010, Aguilera et al. (2011) found only 6% included continuous or hybrid data, and we could only find 9 more recent examples in the literature (web of science search in November 2021 with terms (environmental AND modelling* AND "Bayesian network" AND continuous), with manual sorting of results).

Here, we develop a GBN to forecast seasonally water quality in the western basin of lake Vansjø, a shallow mesotrophic/eutrophic lake in southeast Norway. A number of BN models have previously been applied in the lake (Barton et al., 2008; Couture et al., 2018; Couture et al., 2014; Moe et al., 2019; Moe et al., 2016), but these were all discrete meta-models, i.e. the underlying network nodes were 'response surfaces' summarising a combination of process-based model simulations, expert opinion or data distributions, and the studies were focused on the longer-term impacts of climate, land

use and land management change. Here, the aim was to provide medium-term forecasts to support lake management, by developing a model able to  predict, in spring, water quality for the coming growing season (May – October), including the probability of lying within WFD ecological status classes for TP, chl-a and cyanobacterial. To develop the model we took a data-driven approach: we first use exploratory statistical analyses to identify the main controls on interannual variability in lake water quality, then combine the results of this with domain knowledge to develop the GBN, and finally parameterise it

using 39 years of data. For comparison, we also develop a discrete BN. We then explore the sources of predictability and the importance of weather variables by comparing GBN predictive performance of different model structures within a cross validation scheme, as well as comparing BN predictive ability to a simple benchmark model.

## 2.  Methods and data

### 2.1.     Case study site

Lake Vansjø is a large lake in southeast Norway (59°24′N 10°42′E; 25 m asl), with a highly agricultural catchment by Norwegian standards (15% of the 690 km² catchment is agriculture) and clay- and P-rich soils. The lake has two main basins, Storefjorden in the east (24 km²) and Vanemfjorden in the west (12 km²) (Fig. 1). The largest input is the Hobøl River (catchment area 301 km²), which enters Storefjorden, and then water flows from Storefjorden to Vanemfjorden through a narrow channel (Grepperodfjorden), and from Vanemfjorden through Mosselva and into the Oslo Fjord (Fig. 1).

Over the period 1989-2018, catchment mean annual air temperature was 7.2 °C and annual precipitation was 992 mm yr⁻¹.



Here, we focus on Vanemfjorden, which is shallower (mean depth 3.8 m, max depth 19 m) and more susceptible to eutrophication and cyanobacterial blooms than Storefjorden. Vanemfjorden has a relatively short residence time (0.21 years) and the water column remains oxygenated throughout the year. Vanemfjorden has a long history of eutrophication, and is

usually in WFD 'Moderate' ecological status in relation to mean growing season TP (> 20 μg/l), chl-a (> 10.5 mg/l) and maximum cyanobacteria (> 1.0 mg/l) (Skarbøvik et al., 2021). Vanemfjorden suffers from toxin-producing cyanobacterial blooms and bathing bans were in place during much of the early 2000s (Haande et al., 2011).

The outlet of Vanemfjorden is dammed, and lake water level is regulated for hydropower, recreation, and flood protection.

There is a management opportunity for the dam operators to adjust the water level in advance of an anticipated wet, dry or hot season if problematic water quality was expected, whilst the local catchment management group (Morsa), responsible for WFD implementation, are interested in seasonal water quality forecasts to inform their management plan, in particular preparedness for cyanobacterial blooms.



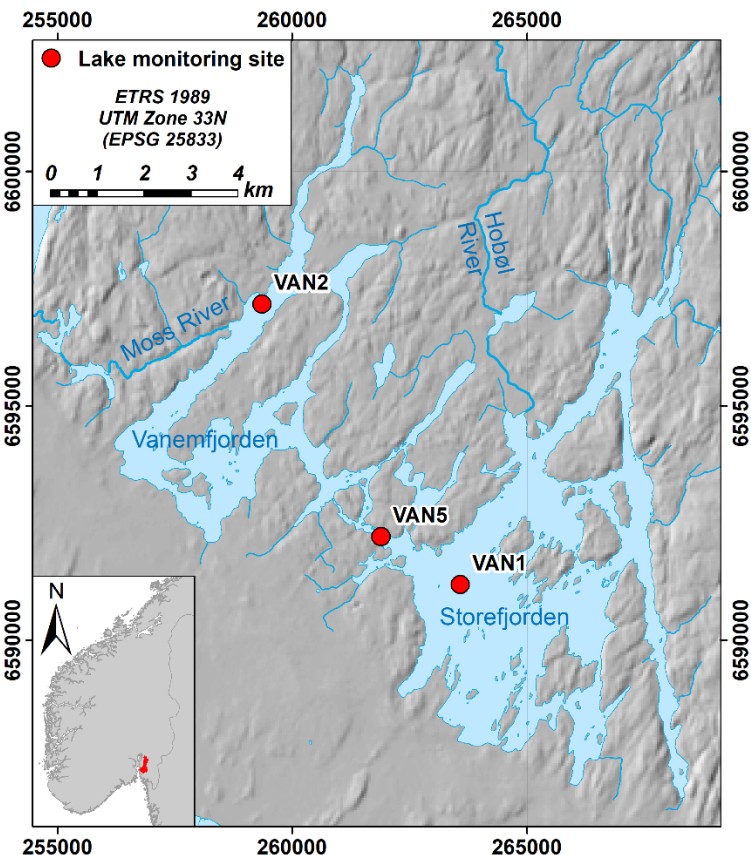

**Figure 1. Lake Vansjø in southeast Norway, showing the two main basins, Vanemfjorden (the study basin) and the larger eastern basin (Storefjorden). The two are connected by a narrow channel. Main NIVA monitoring sites are shown. Here, we use data from Van2.**

## 2.2. Overview of the workflow

The aim was to develop a model to produce probabilistic forecasts, issued in spring of a given year, of expected growing season (May-October) mean concentrations of TP and chl-a and maximum cyanobacteria biovolumes, as used in WFD status classification for Norwegian lakes (Vanndirektivet, 2018). Mean lake colour was also forecast, both because it is of interest for drinking water treatment, and because it may influence algal biomass by affecting nutrient and light conditions (Bergström & Karlsson, 2019; Carpenter et al., 1998).

The model development and assessment workflow consisted of the following steps:



1. *Feature generation*: Data pre-processing and temporal aggregation to derive an array of potential explanatory variables (or features).

2. *Feature selection*: Exploratory statistical analyses to identify key features, using a combination of correlation coefficients, scatterplots and feature importance analysis using regression trees. Process knowledge was used as the final selection criteria.

3. *BN development*: the selected explanatory variables were incorporated into a GBN, using process knowledge to define the structure. Data from the study site were then used to parameterise the model. A discrete BN was also developed for comparison.

4. *BN cross-validation and evaluation*: selection of the most appropriate GBN structure for each target variable, with a particular focus on any added value from including weather variables, and comparison to the discrete BN.

5. *Benchmarking*: Comparison of GBN predictive skill to a simple benchmark model, a seasonal naïve forecaster.

All pre- and post-processing was carried out in the Python programming language. BN development and cross-validation were carried out using the BNLearn R package (Scutari, 2009; Scutari & Ness, 2012). Scripts and data are available in the GitHub repository (see Section 'Code and data availability').

## 2.3.     Data and temporal aggregation

Meteorological, river flow, river chemistry and lake chemistry data were used to derive potential explanatory variables. Precipitation and air temperature were derived from the seNorge 1 km$^2$ gridded data (Lussana et al., 2019), averaged over the whole catchment. Wind speed data were from the met.no monitoring location at Rygge airport, by the southern edge of the lake. River discharge is measured hourly by NVE at Høgfoss and was aggregated to a daily sum. TP concentration data from the Hobøl River at Kure were downloaded from Vannmiljø (https://vannmiljo.miljodirektoratet.no/; last accessed Nov 2021).

Lake water quality data from the surface 0-3 m from monitoring point Van2 (see Fig. 1) were used. TP, chl-a and colour data were downloaded from Vannmiljø whilst cyanobacteria biovolume was provided by NIVA (pers. comm). NIVA colour data was patchy over the period 1998-2007. However, water colour is also monitored by Movar, the local drinking water company, and was obtained for the period 2000-2012 (pers. comm.). Despite different sampling locations and depths (Movar monitoring is in Storefjorden at 20 m depth), the two datasets were highly correlated and from the same distribution. We therefore patched the series together, using NIVA data pre-1999, Movar data from 1999-2012 and NIVA data from 2013. Cyanobacteria monitoring began in 1996, whilst all other variables were monitored since 1980. Prior to 2004, sampling took place 6-8 times a year during May/June to September/October. From 2005, the period changed to mid-April to mid-October, and with higher frequency (fortnightly for cyanobacteria, weekly for other variables between 2005 and 2014 and fortnightly thereafter). The number of samples per growing season therefore varies considerably throughout the period 1980-2018, from



5-10 per year until 2004, increasing to around 25 (TP, chl-a, colour) until 2013, and then decreasing to around 12. Monthly
and seasonal means pre-2005 are therefore based on substantially fewer data points.

Lake TP concentration in Vanemfjorden is fairly constant throughout the whole May-October growing season, and is almost
always in the range 25-40 ug/l. Meanwhile, river TP concentrations are almost always above this, around 40-140 ug/l. Chl-a
and cyanobacteria biovolume tend to peak in July or August. Lake colour is highest in spring and winter and decreases
through summer and autumn.

As the aim was to predict the WFD status class of a number of key water quality parameters, which in Norwegian lakes are
assessed using average or maximum values over the whole growing season (May-Oct) (Solheim et al., 2014). Daily data
were therefore truncated to the growing season (May-Oct) and were aggregated over this period by calculating seasonal
means, sums, counts or maxima. This 6-monthly aggregated data was then used in all subsequent analyses. Time series for
the four lake water quality variables of interest and a number of potential explanatory variables, aggregated over the summer
growing season, are shown in Fig. 2. Interannual variability in TP is low, aside from a general decline since around 2001.
Chl-a is more variable, although longer-term trends still dominate, with an increase until around 1995, high values during
1995-2006, and decreasing thereafter. Cyanobacteria was variable until 2008 and has been low since. There is a step change
increase in lake colour between 1997 and 1999. Lake colour has been increasing across Scandinavia over recent decades, so
this may be real (de Wit et al., 2016), but it may also be due to e.g. a change of labs or methods, but this could not be
confirmed due to a lack of metadata. Some broad-scale trends are also apparent in in the potential explanatory variables.
Growing season mean air temperature is generally between 12 and 14°C, but was somewhat higher after 2005. Mean wind
speed was highest earlier in the period in the 1980s, lowest around 2006-2008, and increased thereafter. This increase over
the last decade appears to be mostly due to a lack of calm wind days, and is observed at other nearby meteorological stations
(e.g. Skarpsborg). Precipitation shows high variability, but was generally lower in the first half of the study period.

Temporal aggregation over the whole growing season, although of WFD-relevance, is coarse and may miss causative
relationships. We therefore also carried out finer-scale aggregation, including: (1) *Algal peaks and pre-peak conditions for*
*explanatory variables*: For each year, we selected peak values for chl-a and cyanobacteria (i.e. maxima). We then calculated,
for each of chl-a and cyanobacteria, means or sums of the potential explanatory variables over 14, 30, 60 and 90 days pre-
peak. By ensuring that the potential explanatory variables only included data from *before* the observed algal peak, this
aggregation method should have more power to identify causative relationships, whilst still focusing on factors controlling
inter-annual variation. (2) *Monthly aggregation*. A repeat of the exploratory statistical analysis (Section 2.5) using monthly
data includes both within and between year variability.



**Figure 2. Time series of growing season (May-Oct) mean concentrations of lake chl-a (mg/l), total phosphorus (TP; µg/l), colour (mg Pt/l), wind speed (m/s), air temperature (°C) and Hobøl River TP concentration (µg/l); seasonal maxima of cyanobacteria biovolume (mg/l); and seasonal sums of rainfall (mm) and discharge (Q, ×10⁶ m³) for the western basin (Vanemfjorden) of Lake Vansjø.**

## 2.4.   Feature generation

Using process knowledge and the literature as guidance, we used the daily data to generate a set of potential explanatory variables (or features, in machine learning parlance). These included weather-related features, features relating to the delivery of water and TP from the catchment, and inter-connections between the dependent variables. Feature generation was



largely limited to variables that could be measured or potentially forecast (e.g. using a seasonal climate forecast) at the time when the forecast would be issued in spring of a given year. Some potentially relevant features, e.g. water quality in the eastern lake basin, water temperature or water column stability indices, were not included, as they would need to be included as latent variables in the GBN, increasing its complexity. Features were generated for the current growing season, the

previous year's growing season and the previous winter (the 6 month period prior to the current season), to take into account the potential influence of previous conditions. Overall, we generated up to 33 potential explanatory variables, depending on the response variable. Features considered for all target variables are given in Table 1 and additional features, specific to a given target variable, are given in Table 2.

The date range for the derived features was 1981 – 2018. Depending on the number of years with missing data, this gave 39 years of data for TP and chl-a, 36 for lake colour and 24 for cyanobacteria for model training and validation.

**Table 1: Features generated for all target variables. All were repeated for the previous 6-month winter period. Wind percentiles relate to the period 1980-2018.**

| Feature name | Description |
|---|---|
| Pptn | Precipitation sum (mm) |
| Rain_day | Count of days with precipitation (daily precipitation $\geq$ 1mm) |
| Pptn_intense | Count of days with intense precipitation (daily precipitation $\geq$ 10 mm) |
| Q | Inflow discharge sum ($10^6$ m$^3$) |
| Temp_subzero_winter | Count of days the previous winter with daily mean temperature < 0 °C |
| Temp | Mean of daily mean temperature (°C) |
| Temp_prevSummer | Mean air temperature the previous summer (May-Oct; °C) |
| Rel_res_time | Relative residence time (estimated as 1/Q) |
| Wind_speed | Mean of daily mean wind speed (m/s) |
| Wind_under_P20 | Count of days when daily mean wind speed < 20$^{th}$ percentile (2.0 m/s) |
| Wind_under_P40 | Count of days when daily mean wind speed < 40$^{th}$ percentile (2.9 m/s) |
| Wind_over_P60 | Count of days when daily mean wind speed > 60$^{th}$ percentile (3.8 m/s) |
| Wind_over_P80 | Count of days when daily mean wind speed > 80$^{th}$ percentile (4.8 m/s) |






**Table 2: Additional features, specific to a given target variable. From TP onwards these are cumulative as you go down the table, so that additional features for chl-a, for example, are features listed for both TP and chl-a.**

| Target variable | Feature name | Description |
|---|---|---|
| colour | colour_prevSummer | Mean lake colour the previous summer (mg Pt/l) |
| TP | TP_catch | Mean TP concentration in the Hobøl River (µg/l) |
| | TP_prevSummer | Mean lake TP concentration the previous summer (µg/l) |
| chl-a | TP | Mean lake TP concentration (µg/l) |
| | chl-a_prevSummer | Mean chl-a concentration the previous summer (mg/l) |
| cyano | chl-a | Mean chl-a concentration (mg/l) |
| | cyano_prevSummer | Maximum cyanobacterial biovolume the previous summer (mg/l) |
| | colour | Mean lake colour (mg Pt/l) |

## 2.5. Feature selection

Having generated a list of potential explanatory variables for each target variable, we then carried out exploratory statistical
analyses to select the features to include in the GBN, using a combination of:

1. *Ranked correlation coefficients*: As a first screening, we used ranked absolute correlation coefficients to highlight potentially important features for each dependent variable.

2. *Feature importance*: We also used a more formal machine learning approach to assess feature importance, using random forests implemented using the Scikit-Learn python package (Pedregosa et al., 2011). Random forests use bootstrapping
to partition the data used by each tree, and data not included in each bootstrap sample are used to perform internal validation. We used the "out-of-bag" (OOB) score and importance scores to rank feature importance. We used recursive feature elimination to try to find the best random forest regressor model using subsets of the available features. This is similar to stepwise regression, but uses cross validation to avoid overfitting, rather than traditional significance testing, and in this case we used out-of-sample $R^2$ to measure performance. Random forests have a number of hyperparameters
that can be tuned to improve performance. The most important are the number of trees in the forest (n_estimators) and the size of the random subsets of features to consider when splitting a node (max_features). We selected values for these by plotting the OOB error rate (1 – OOB Score) as a function of n_estimators for various choices of max_features.

3. *Visual evaluation of relationships*: for each target variable, scatterplot matrices were used for a visual check of whether relationships appeared to be linear and for independence between explanatory variables (required for unconnected nodes
in a BN).

4. *Process understanding*: Finally, we considered whether there were plausible physical mechanisms underlying the relationships.

## 2.6. Bayesian network development and use in prediction

We first defined the BN structure manually, using results of the exploratory feature selection and process-knowledge, to
ensure realistic causative relationships between nodes.





As mentioned in the introduction, Gaussian Bayesian networks (GBNs) are a powerful class of continuous BNs in which all nodes are continuous and conditional probability distributions (CPDs) are linear Gaussians, which together define a joint Gaussian. Parent nodes therefore have simple normal distributions with mean $\mu$ and variance $\sigma^2$. Gaussian CPDs of child nodes have a mean which is a linear combination of the parent nodes (with intercept $\beta_0$ and coefficients $\beta_n$). To meet the normality requirement of GBNs, we transformed the cyanobacteria data, which were right skewed with many zeros, by applying a box cox transformation ($y^* = (y^L - 1)/L$ with a lambda $L$ of 0.1 to give a fairly symmetrical distribution. Normality tests for all variables showed high p values ($>0.2$) for all but lake colour ($p = 0.03$) and transformed cyanobacteria ($p = 0.05$). A step change in lake colour is seen around 1998 (Fig. 2) suggesting the distribution of lake colour may be bimodal. The normality assumption was therefore not invalidated at a 1% significance level, but would have been at a 5% level. This weakness should be taken into consideration when interpreting results. Coefficients were then derived for the CPDs at each node using maximum likelihood estimation.

BNs can be used for prediction, our primary aim, by calculating a probability distribution over the variable whose value we would like to know, given information (evidence) we have about some other variables. Predictions obtained using GBNs contain a mean and a variance, and here we computed predictions in BNLearn by averaging likelihood weighting simulations using a subset of nodes as evidence. The predicted value is then the expected value of the conditional distribution. We chose the evidence nodes based on those nodes which would be updated whenever a forecast was produced, using historic data or future forecasts (i.e. observed water quality from the previous summer or forecasted meteorological conditions).

A particular advantage of using GBNs is that they can be used not only to predict a given variable, but they also specify the posterior distribution of the response variable. This in turn can be used to determine the risk that the response variable passes a certain threshold, which may be particularly useful where the interest may be the probability of failing to meet certain environmental thresholds. As well as predicting absolute values, we therefore also estimated the probable WFD-relevant ecological status class for each variable. We used a single WFD-relevant threshold per variable, i.e. a binary classification, as follows:

- *TP*: almost all TP observations are in the Moderate WFD status class, so used a threshold of 29.5 µg/l to classify TP as 'Lower moderate' or 'Upper moderate'.
- *chl-a*: Few data were under the Good/Moderate boundary of 10.5 mg/l, so we used the Moderate/Poor boundary of 20.0 mg/l to classify chl-a as either 'Moderate or better' or 'Poor or worse'.
- *Cyanobacteria*: the majority of observations were below the Moderate/Poor threshold (2.0 mg/l), so we used the 1.0 mg/l Good/Moderate boundary to classify status as 'Moderate or worse' or 'Good or better'.
- *Colour*: There were no obvious management-relevant thresholds to apply, so we used the 66th percentile (48 mg Pt/l) to classified colour as 'High' or 'Low'.





Finally, we developed a discrete BN, for comparison with the GBN. To do this, we first discretized the data, opting again for just two classes per variable, given the small sample size for fitting conditional probability tables (CPTs). We used the bounds mentioned above for TP, chl-a, cyanobacteria and colour for the current season. For all other features (including lake
observations from the previous summer), we used regression trees to discretize, picking the topmost division. For wind speed, this resulted in highly unbalanced class sizes, so we instead used the median. We then fitted the CPTs using BNLearn's 'bayes' method, a classic Bayesian posterior estimator with a uniform prior.

## 2.7.     BN validation and assessment

We then explored the most appropriate GBN model structure and assessed its predictive performance using three methods:
(1) cross validation, carried out on several parts of the network separately and including comparison to the discrete BN; (2) goodness of fit of the whole network compared to observations; and (3) comparison to a simple benchmark model.

### 2.7.1.     Cross validation

The ability to carry out cross validation (CV) is a great benefit of using BNLearn compared to many graphical BN packages, as it is possible to assess the expected performance of the network for out-of-sample prediction, and to compare different
structures to robustly assess whether certain nodes and arcs are providing worthwhile predictive power. Here, we used CV to compare the predictive performance of GBNs with and without meteorological nodes, and to compare the GBN and the discrete BN. We used leave-one-out cross validation, which produces unbiased skill score estimates and is well suited when sample sizes are small (e.g. Wong, 2015). As the main aim was prediction, we used posterior predictive correlation (reported as $R^2$) and mean square error (MSE) as the network skill scores, and repeated the procedure for each dependent variable. We
used the classification error  (the proportion of the time the classification was incorrect) as the skill score for the discrete BN, and we calculated this manually for the GBN for comparison. Model predictions were derived from a specified set of nodes using likelihood weighting to obtain Bayesian posterior estimates. The cross validation is stochastic and was run a default 20 times and the mean of skill scores were calculated.

Cross validation requires complete data for all variables and years. For most variables there were few gaps, and we filled up to one year gaps by interpolation or backward/forward filling. However, cyanobacteria was only measured in the lake from 1996, whilst other variables were measured from 1980. Rather than dropping all data prior to 1996, which would result in a large loss of training data for TP, chl-a and colour, we instead split the network into a number of smaller networks for the target variables, and cross validated each of these in turn (see Section 3.3.1).



### 2.7.2. Goodness-of-fit of the whole network

Splitting the BN up into smaller sub-networks is likely to result in a loss of predictive power, so cross validation could not be used to assess the expected predictive performance of the whole network at performing out of sample forecasts. Instead, we also assessed performance of the whole network, trained on all data, by simply calculating goodness-of-fit of predictions against observations, and once again using a GBN with and without weather nodes. To assess skill, we used the same correlation and MSE statistics as during cross validation, as well as bias (mean of (predicted – observed)). We also calculated two categorical skill scores, which reflect how well the WFD status class was predicted: Matthew's correlation coefficient (MCC), which is in the range 0 (no skill) to 1 (perfect skill), and is an informative and truthful score for evaluating binary classifiers (Chicco & Jurman, 2020), and the classification error. As the training and evaluation data were the same in this case, this may produce an optimistic assessment of model performance.

### 2.7.3. Comparison to a benchmark model

Some extremely simple forecasting methods can be highly effective. As a final test, we compared predictive performance of the GBN to a simple benchmark model, a seasonal naïve forecast (Hyndman & Athanasopoulos, 2018). In this case, the seasonal naïve forecast for the current growing season is simply the observed value from the previous year's growing season.

## 3. Results

### 3.1. Feature selection

#### 3.1.1. Feature selection using 6-monthly temporal aggregation

For lake TP concentration, the strongest correlation was with TP concentration from the previous growing season (Table 3). Otherwise, the only correlation coefficients above 0.2 were with wind features, the strongest being a negative relationship with number of calm days (wind_under_P20). These two features were also selected as most important in the feature importance analysis; the rest all had importance scores under 0.1 (Table 4). A regression tree model with just these two features had an "out-of-bag" (OOB) score of 0.35, only a little lower than when all features were included (Table 4). No features relating to delivery of P to the lake (e.g. discharge or river TP concentration) came out as being important. Temporal autocorrelation in lake TP concentration is highly plausible. It is however less clear whether the negative correlation with wind speed is causative. We might expect windier conditions to decrease stratification and increase mixing and sediment resuspension, and result in higher rather than lower TP concentrations (Hanlon, 1999). Meanwhile, a positive relationship was seen between the previous summer's TP and wind the following winter (Fig. 3) which, together with results of analyses using monthly aggregated data (Section 3.1.2), suggest the relationship may not be causative. Wind was not therefore selected for TP.





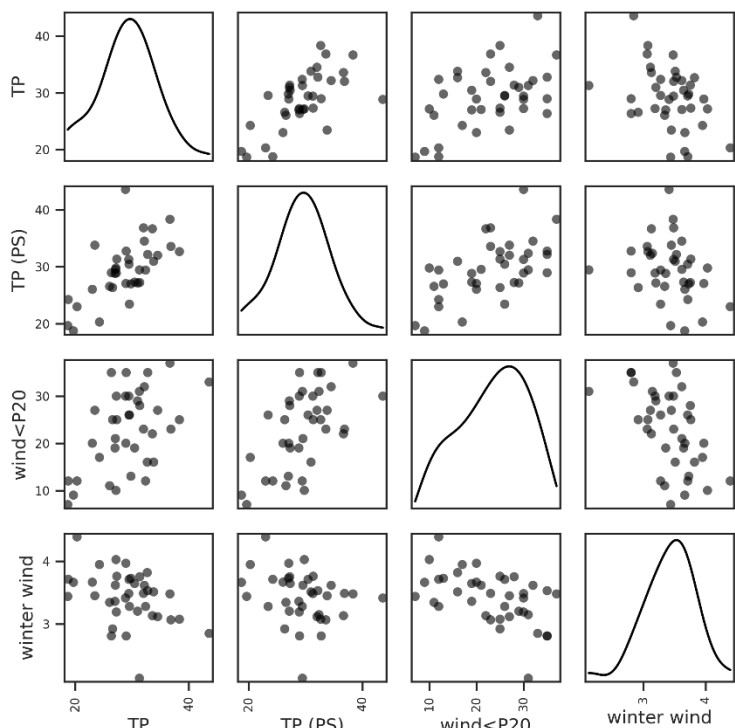

**Figure 3: Relationships between seasonal mean lake TP concentration (μg/l), TP observed the previous summer (PS), number of days when daily mean wind speed < 20th percentile (wind<P20), and mean winter (Nov-April) wind speed (m/s).**

**Table 3: Pearson's correlation coefficients (R) for the four target variables (only |R| > 0.40 are shown).**

| TP | | Chl-a | | Cyano | | Colour | |
|---|---|---|---|---|---|---|---|
| Variable | R | Variable | R | Variable | R | Variable | R |
| TP_prevSummer | 0.65 | chl-a_prevSummer | 0.65 | chl-a | 0.77 | colour_prevSummer | 0.85 |
| wind < P20 | 0.51 | TP | 0.58 | TP | 0.58 | pptn | 0.53 |
| wind < P20_winter | 0.44 | wind < P40 | 0.41 | chl-a_prevSummer | 0.56 | pptn_intense | 0.46 |
| wind_speed_winter | -0.40 | wind > P80 | -0.49 | cyano_prevSummer | 0.55 | Q | 0.45 |
| | | wind speed | -0.51 | TP_prevSummer | 0.49 | temp_prevSummer | 0.43 |
| | | wind > P60 | -0.51 | colour | -0.44 | wind > P60 | -0.45 |
| | | | | colour_prevSummer | -0.50 | wind_speed | -0.46 |
| | | | | | | wind > P80 | -0.47 |





**Table 4: Summary of feature importance analysis, and feature importance scores and OOB score for the proposed GBN feature set. OOB is the out-of-bag score. See Tables 1 and 2 for a description of the features.**

| Target variable | Feature subset | Feature | Importance scores | OOB |
|---|---|---|---|---|
| TP | All | TP_prevsummer | 0.2 | 0.40 |
| | | wind_under_P20 | 0.14 | |
| | | All others | <0.1 | |
| | Top 1 (for GBN) | TP_prevsummer | 1 | 0.10 |
| | Top 2 | TP_prevsummer | 0.60 | 0.35 |
| | | wind_under_P20 | 0.40 | |
| chl-a | All | chl-a_prevsummer | 0.29 | 0.48 |
| | | TP | 0.21 | |
| | | wind_speed | 0.05 | |
| | | all others | <0.05 | |
| | Optimum | chl-a_prevsummer | 1 | 0.36 |
| | Proposed for GBN | chl-a_prevsummer | 0.41 | 0.49 |
| | | TP | 0.34 | |
| | | wind_speed | 0.24 | |
| cyano | All | chl-a | 0.18 | 0.37 |
| | | colour | 0.08 | |
| | | All others | <0.07 | |
| | Optimum | chl-a | 1 | 0.35 |
| | Proposed for GBN | chl-a | 0.62 | 0.54 |
| | | colour | 0.38 | |
| colour | All | colour_prevsummer | 0.73 | 0.64 |
| | | All others | <0.06 | |
| | Optimum | colour_prevsummer | 0.79 | 0.66 |
| | | wind_under_P20 | 0.12 | |
| | | pptn | 0.09 | |
| | Proposed for GBN | colour_prevsummer | 0.85 | 0.57 |
| | | pptn | 0.15 | |

For chl-a, strongest correlations were with chl-a the previous summer and lake TP concentration (Table 3). Otherwise, the only correlation coefficients above 0.4 were with wind-related features, e.g. a negative relationship with mean wind speed. This was partly supported by the feature importance analysis, although wind variables were not picked out as being important even though the highest OOB score included a wind feature (Table 4). We therefore selected previous summer's chl-a and lake TP as key predictors for chl-a, and there are plausible mechanisms that can underpin these relationships. It was less clear whether to include wind. Windier summer weather can cause less stable lake stratification and lower chl-a concentrations (Huber et al., 2012; Yang et al., 2016), so there is a plausible mechanism. Including it could also help improve the TP forecast, through the TP - chl-a link. We therefore decided to include wind to start with, but to investigate its importance through cross validation. As we will see, temperature exerted an important control on within-year changes in chl-a (see Section 3.1.2), but there was no evidence that years with higher summer air temperature were associated with higher mean chl-a concentration (Fig. 4).



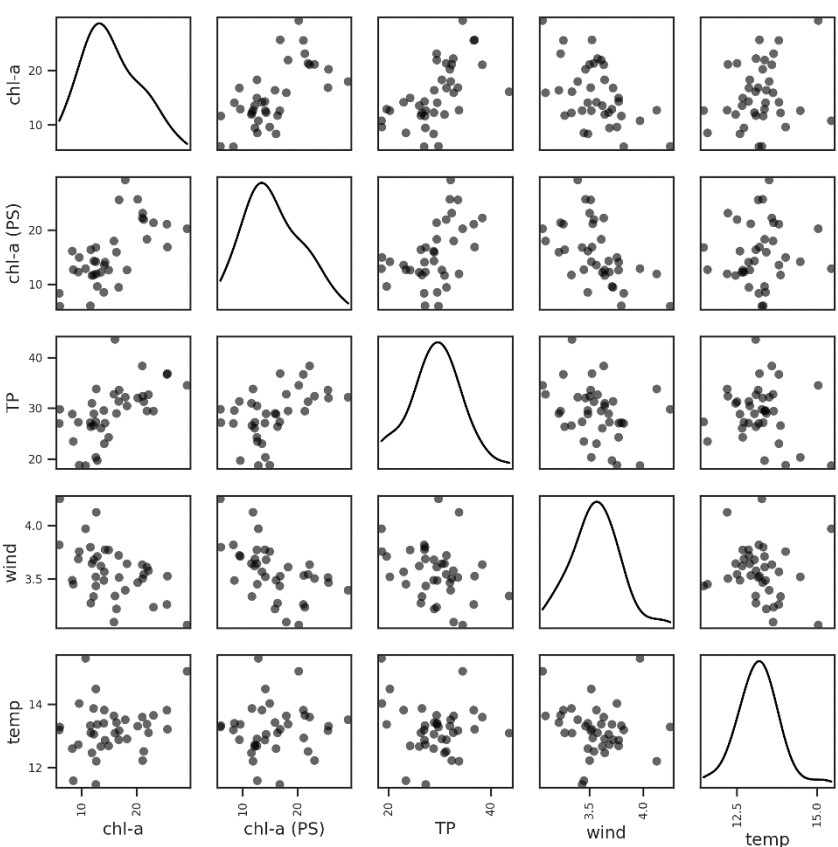

**Figure 4: Relationships between seasonal mean chl-a (mg/l) and chl-a from the previous summer (PS), seasonal mean TP (µg/l), wind speed (m/s) and air temperature (°C).**

For cyanobacteria, by far the strongest correlation was with lake chl-a, although a number of other correlations were present (Table 3, Fig. 5) including with lake TP concentration and colour, and concentrations of chl-a, TP and cyanobacteria the previous summer. Cyanobacteria was less correlated with wind-related variables than chl-a, and there were no (or even negative) correlations with seasonal air temperature. Feature importance analysis highlighted chl-a as the most important variable (Table 4), and highest OOB values (0.54) were obtained using just chl-a and lake colour. Given strong correlations between chl-a and TP and chl-a_prevSummer, these latter two features were dropped, and we selected just chl-a and colour as key explanatory variables for cyanobacteria. The relationship between lake colour and cyanobacteria is plausible, as an increase in organic matter can affect lake algal communities by reducing light availability and the availability of nutrients such as nitrogen, P and iron, as they become bound to the organic matter (Nagai et al., 2006). Senar et al. (2021) found that above DOC concentrations of 8-12 mg/l, similar to those observed in lake Vanemfjorden (7-10 mg/l over the period 1996-2018), cyanobacteria became replaced by mixotrophic species as lake colour increased.





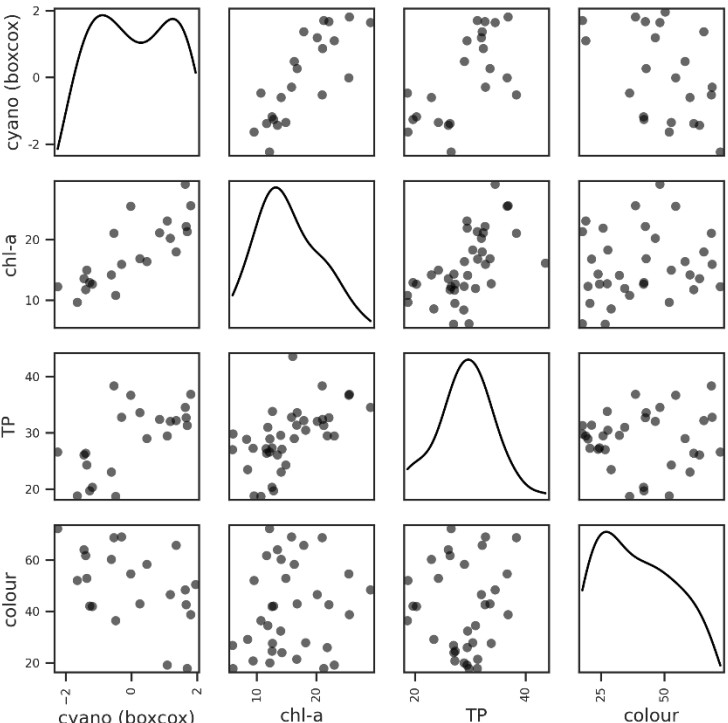

**Figure 5: Relationships between maximum seasonal cyanobacteria biovolume (original units mg/l; Box Cox transformed) and seasonal means in lake chl-a (mg/l), TP (µg/l) and colour (mg Pt/l).**

Lake colour was very strongly correlated with the previous summer's colour (R = 0.85), and, probably because of this, the OOB score for lake colour was the highest of all the target variables (0.66). Colour was moderately correlated with factors

relating to catchment delivery (Table 3, Fig. 6). Feature importance analysis resulted in an optimum at 3 features, including calm wind days and precipitation, although their importance scores were low compared to the previous summer's colour (Table 4). Whilst it is clear that higher rainfall can lead to higher catchment delivery of organic matter, and therefore higher lake colour, once again it is less clear whether wind should be included as an explanatory variable. Lake colour is relatively uniform throughout the water column in Vansjø, and so the impact of wind on lake stratification should be minimal. In

addition, there was a negative relationship between wind and rain (Fig. 6). We therefore decided to just select previous summer's colour and precipitation as explanatory variables for colour.



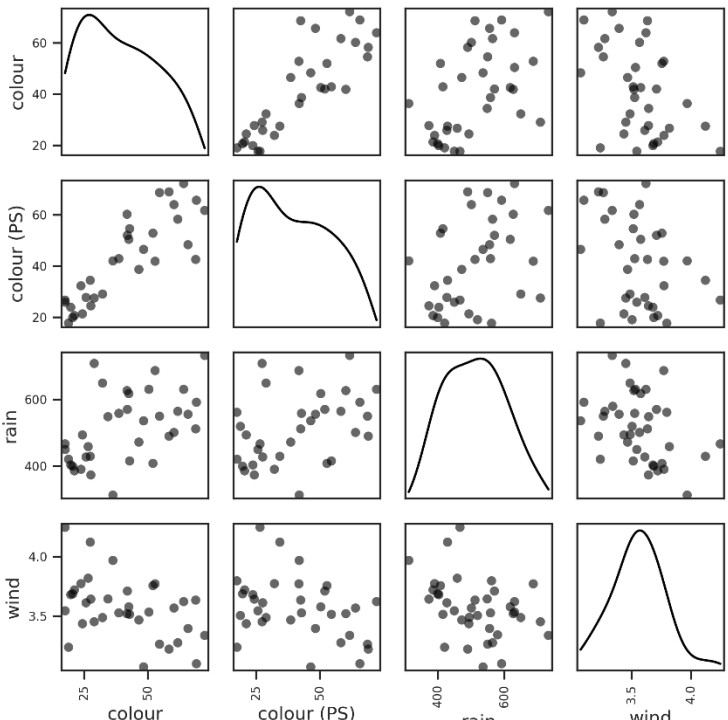

**Figure 6: Relationships between seasonal mean lake colour (mg Pt/l) and colour the previous summer (PS), seasonal precipitation sum (mm) and mean wind speed (m/s).**

In summary, the following features were selected for BN development for the four target variables:

- TP: lake TP concentration from the previous summer
- chl-a: chl-a from the previous summer, lake TP concentration, wind speed
- cyanobacteria: lake chl-a and colour
- colour: colour from the previous summer, precipitation

### 3.1.2. Exploratory statistical analyses using finer temporal aggregation

*a) Algal peaks and pre-peak conditions for the explanatory variables*

We then looked for relationships between seasonal maxima of chl-a and cyanobacteria, and potential explanatory variables aggregated over $n$ days ($n = 14, 30, 60, 90$) before the maxima were observed (Section 2.3). For chl-a, strongest relationships were seen with wind speed and related variables and lake TP concentration (Table 5), as in the analysis using 6-monthly aggregation. No other weather variables were important. For cyanobacteria, strongest correlations were with lake TP and chl-a concentrations, and there was also a relationship with lake colour, as in the 6-monthly analysis. In contrast to the whole-seasonal analysis, relationships between cyanobacteria and variables relating to wetness and flow were seen for some




temporal aggregation windows, suggesting that the larger the rainfall and river discharge (and the shorter the residence time) over the preceding 30-60 days, the lower the cyanobacterial biomass. Overall, this analysis using a shorter and more causally-plausible temporal aggregation resulted in very similar features being selected as being important as in the whole-season aggregation. The exception was that hydrology and residence time may play more of a role in cyanobacteria bloom

development than is acknowledged in the whole-season GBN.

**Table 5: Pearson's R correlation coefficients between seasonal maxima of chl-a and cyanobacteria and potential explanatory variables aggregated (mean or sum) over *n* days before the algal peak occurred. For clarity, only |R| > 0.20 are shown for chl-a and |R| > 0.30 for cyanobacteria.**

| Variable | Temporal aggregation over *n* days pre-peak | | | | | | | |
|---|---|---|---|---|---|---|---|---|
| | n = 14 | | n = 30 | | n = 60 | | n = 90 | |
| Chl-a | wind_speed | -0.35 | wind_speed | -0.24 | wind > P80 | -0.31 | wind > P80 | -0.32 |
| | wind > P80 | -0.32 | wind > P80 | -0.22 | wind_speed | -0.25 | wind_speed | -0.23 |
| | | | | | wind > P60 | -0.23 | | |
| | TP | 0.21 | TP | 0.21 | TP | 0.34 | TP | 0.36 |
| | wind < P40 | 0.23 | wind < P20 | 0.23 | | | | |
| | wind < P20 | 0.27 | | | | | | |
| Cyano | colour | -0.33 | rain_day | -0.41 | rain_day | -0.45 | colour | -0.41 |
| | Q | -0.31 | pptn | -0.36 | pptn | -0.39 | | |
| | | | Q | -0.33 | colour | -0.38 | | |
| | | | colour | -0.33 | | | | |
| | chl-a | 0.48 | chl-a | 0.54 | chl-a | 0.48 | TP | 0.51 |
| | TP | 0.71 | TP | 0.63 | TP | 0.61 | chl-a | 0.55 |

*b)   Monthly aggregation*

For all variables, strongest relationships were with values observed the previous month(s), and there were strong correlations between values observed the previous summer. As well as this strong temporal auto-correlation, potentially important relationships included:

• TP: weak relationships with wind, as in the 6-monthly analysis. For example, the calmer the previous 2-6 months, the higher the TP (R = 0.26 or less, depending on the lag), and the windier the previous winter or 6 months, the lower the TP (R = -0.2). That stronger relationships were seen between TP and wind over the previous ≥ 2 months, rather than the previous or current month, is suspicious given that wind would likely have an immediate and relatively short-lived effect on TP via water column mixing, and supports our suspicion that the relationship is not

causative. Relationships with all other variables were weak (R < 0.16).





- Chl-a: strongest relationships were with air temperature from the current month (R = 0.54) and related lagged variables, discharge (R = -0.39), lake TP concentration (R = 0.32) and calm wind days (R = -0.33).
- Cyanobacteria: strongest relationships were with chl-a concentration (R = 0.72), lake colour (R = -0.55), winter wind (R of 0.5 or lower, depending on the wind quantile), and air temperature from the previous month (R = 0.41).

Overall, many of the same variables which were important in explaining inter-annual differences were highlighted as being important in this monthly analysis. However, a key difference is the appearance of a strong relationships between air temperature and chl-a concentration, as discussed further in Section 4.1.

### 3.2. Gaussian Bayesian network development

#### 3.2.1. BN structure and GBN parameters

The key relationships highlighted (Section 3.1) were then used to develop the BN structure, which is shown, together with fitted coefficients for the GBN, in Fig. 7. For parent nodes, coefficients define normal distributions with mean $\beta_0$ and variance $\sigma^2$. Child nodes are linear combinations of the parent nodes with intercept $\beta_0$, coefficients $\beta_n$ and variance $\sigma^2$. Fitted coefficients for the Gaussian BN were all credible, matching the expected relationships between variables seen in the exploratory data analysis.





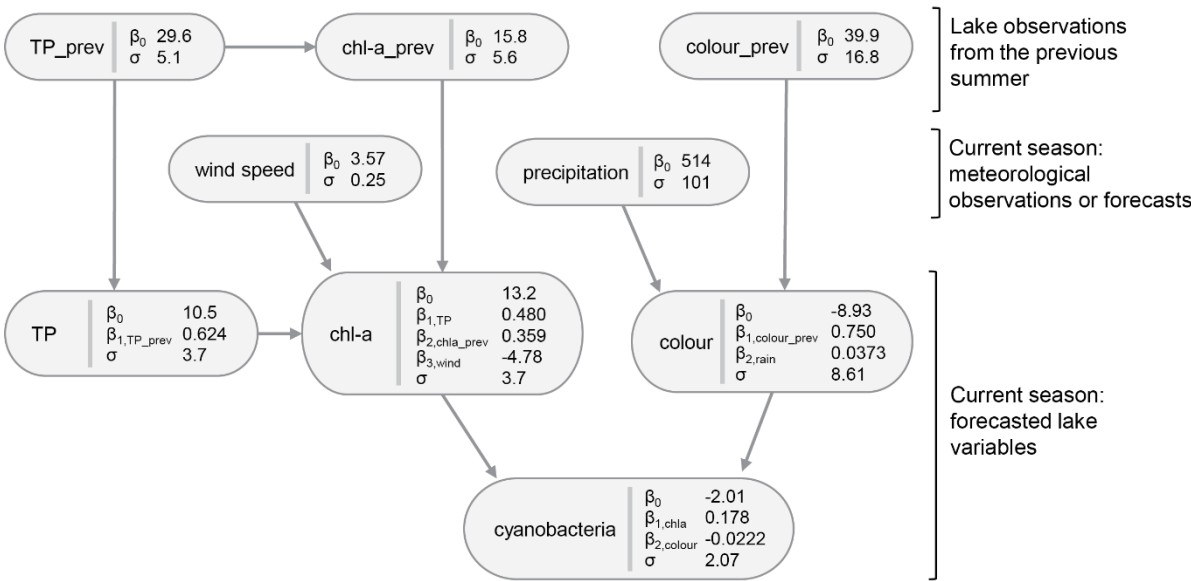

**Figure 7: GBN structure and parameters defining the CPDs at each node. Units for standard deviations (σ) and intercepts (β₀) are the same as the original data aside from cyanobacteria, where a box cox transformation was used. Wind speed is the seasonal mean (m/s) and precipitation is the seasonal sum (mm).**

### 3.2.2. Fitted discrete BN

In contrast to the GBN, the fitted CPTs for the discrete network, using the same structure as the GBN (Fig. 7), did a mixed job of representing the expected relationships between variables. For example, in the fitted probabilities for the chl-a node (Table 6) we see that high wind speed is associated with a greater probability of having high chl-a when the previous summer's chl-a is high and TP is high. This is the opposite effect to that expected (we saw a negative relationship between chl-a and wind). Removing wind from the discrete BN did not fix the problem, as then the marginal probabilities for chl-a did not respond as expected to changing TP. For example, changing TP from low to high corresponded with a decrease in the probability of high chl-a (from 0.94 to 0.74), given high previous summer chl-a. In reality we would always expect a positive (or no) relationship between TP and chl-a. Similar problems were found with cyanobacteria and colour. These are likely artefacts, given low sample sizes for training.





**Table 6: conditional probability table for chl-a, fitted for a discrete version of the BN shown in Fig. 7. Probabilities which do not follow the expected physical response are highlighted. Continuous values were discretized into 'Low' or 'High' classes as described in Section 2.6.**

| Previous summer's chl-a | Wind speed | TP | chl-a L | chl-a H |
|---|---|---|---|---|
| L | L | L | 0.991 | 0.009 |
| | | H | 0.944 | 0.056 |
| | H | L | 0.994 | 0.006 |
| | | H | 0.989 | 0.012 |
| H | L | L | 0.056 | 0.944 |
| | | H | **0.302** | **0.698** |
| | H | L | 0.056 | 0.944 |
| | | H | **0.029** | **0.971** |

### 3.3. GBN validation and assessment

We then explored the most appropriate GBN model structure and assessed its predictive performance using: (1) cross validation to determine the most suitable model structure, including a comparison to the discrete BN; (2) goodness of fit of the whole network compared to observations; and (3) comparison to a simple benchmark model.

### 3.3.1. Cross validation using sub-sets of the network

As mentioned in Section 2.7, cross validation (CV) requires complete data for all variables and years. Given that cyanobacteria was only monitored since 1996, to avoid a large loss of training and evaluation data for TP, chl-a and colour, we split the GBN up into smaller sub-networks before performing cross validation for each target node separately, as follows:

1. TP and chl-a: drop cyanobacteria, colour, previous summer's colour and rain nodes from the BN, and use the whole 1981-2018 period in cross validation.
2. Colour: as colour was linked to the network through cyanobacteria, we had to drop all nodes aside from colour and its parent nodes to be able to include the full period 1981-2018.
3. Cyanobacteria: use the whole network, but only data from 1997.

Cross validation results comparing the classification error of the GBN and the discrete BN are shown in Table 7. We might expect the discrete BN, which was fitted to discrete data, to do a better job of predicting the water quality class than the GBN. However, this was only the case for chl-a and colour.

Predictive performance of the GBN with and without weather nodes is also shown in Table 7, and we can see that lake colour was the only variable for which model performance was a little better when meteorological variables were included, although the gains were marginal. For chl-a, performance was similar with or without weather nodes, and it was identical for





TP. For cyanobacteria, performance was slightly better without meteorological variability, and further investigation showed that this was because of the wind – chl-a relationship. When the wind speed node was dropped, the model skill was as good as when dropping all meteorological variables. Overall therefore, there was a small benefit to keeping precipitation, but no reason to keep wind in the GBN.

**Table 7: Mean predictive performance of different BN structures, including the GBN with and without weather nodes and a discrete BN, assessed through cross validation. Note that the BNs used to make predictions for each target variable were sub-sets of the whole BN shown in Fig. 7 for all but cyanobacteria, to make the most of all available data (see text). R: Pearson's correlation coefficient; RMSE: root mean square error; CE: classification error; GBN: Gaussian Bayesian network.**

| Variable | BN type | Met included? | $R^2$ | RMSE | CE (%) |
|----------|---------|---------------|-------|------|--------|
| TP | GBN | Y | 0.33 | 3.96 | 33 |
| TP | | N | 0.33 | 3.96 | 33 |
| TP | discrete | Y | | | 40 |
| chl-a | GBN | Y | 0.30 | 4.76 | 34 |
| chl-a | | N | 0.29 | 4.76 | 32 |
| chl-a | discrete | Y | | | 8 |
| colour | GBN | Y | 0.72 | 8.78 | 24 |
| colour | | N | 0.68 | 9.35 | 24 |
| colour | discrete | Y | | | 15 |
| cyano | GBN | Y | 0.40 | 1.00 | 15 |
| cyano | | N | 0.46 | 0.96 | 14 |
| cyano | discrete | Y | | | 23 |

### 3.3.2. Goodness-of-fit of the whole network

Model performance assessed by comparing the predictions made using the whole network to observations is shown in Table 8. Performance was best for lake colour, similar for lake TP and chl-a, and slightly lower for cyanobacteria. The same general lack of sensitivity to weather nodes, or for cyanobacteria slightly worse predictive skill when they were included, was seen here as in the CV results, and considering additional model performance measures such as bias and classification skill (Table 8, Fig. 8).





**Table 8: Model performance for GBNs with weather nodes (BN-met) and without weather nodes (BN-nomet), fit using the whole historic period (no cross validation), using the whole BN rather than sub-sets of nodes. Performance of the seasonal naïve forecast is also shown. MCC and classification error reflect classifier skill, whilst the rest reflect how well the mean predicted values matched observations. Abbreviations: RMSE: root mean square error, MCC: Matthew's correlation coefficient.**

| Variable | Model | $R^2$ | RMSE | Bias | MCC | Classification error (%) |
|---|---|---|---|---|---|---|
| TP | naïve | 0.40 | 4.39 | 0.49 | 0.18 | 41 |
| | BN-met | 0.42 | 3.67 | -0.04 | 0.34 | 32 |
| | BN-no met | 0.42 | 3.68 | -0.07 | 0.34 | 32 |
| chl-a | naïve | 0.42 | 4.60 | 0.06 | 0.71 | 11 |
| | BN-met | 0.39 | 4.38 | -0.08 | 0.23 | 27 |
| | BN-no met | 0.37 | 4.44 | -0.06 | 0.18 | 27 |
| colour | naïve | 0.72 | 9.21 | 0.85 | 0.55 | 21 |
| | BN-met | 0.75 | 8.39 | -0.51 | 0.37 | 29 |
| | BN-no met | 0.71 | 9.05 | -0.75 | 0.44 | 26 |
| cyano | naïve | 0.32 | 1.76 | 0.18 | 0.57 | 22 |
| | BN-met | 0.35 | 1.79 | -0.82 | 0.74 | 13 |
| | BN-no met | 0.37 | 1.76 | -0.81 | 0.74 | 13 |

### 3.3.3.   BN predictions compared to a benchmark model

Model performance was then compared to the performance of a seasonal naïve forecaster (Table 8, Fig. 8). For TP, the GBN performed slightly better than the naïve forecaster for all performance statistics, in particular RMSE and bias. Similarly for lake colour and cyanobacteria, the GBN performed better than or comparably to the naïve forecaster, the only exception being that the naïve forecaster produced less biased cyanobacteria predictions. This bias is clear in the BN predictions on Fig. 8, and is likely due to the box-cox transformation used when fitting the BN. Although the GBN was a better cyanobacteria classifier than the naïve forecaster, it's clear on Fig. 8 that, had the WFD-relevant threshold been set at 2 instead of 1 mg/l, the naïve forecaster would have been better. For chl-a, by contrast, the naïve forecaster performed slightly better than the GBN, although this varied among performance statistics.



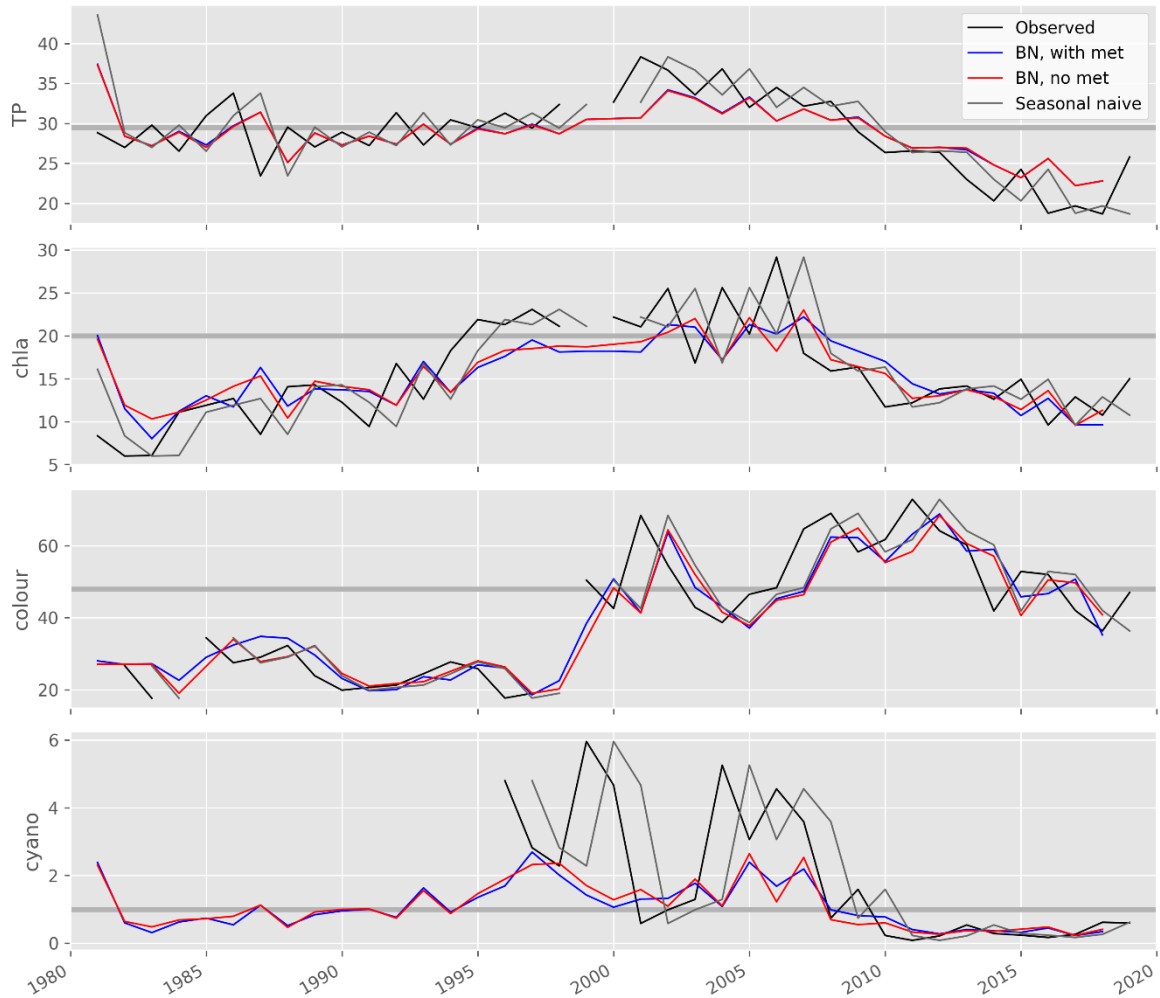

**Figure 8: Observed and predicted (mean) lake water quality variables, including predictions from a range of models: BN with weather variables, BN without weather variables and a seasonal naïve forecaster. Horizontal grey lines show the thresholds used to discretize predictions into two WFD-relevant classes (units: colour: mg Pt/l, TP: µg/l, chl-a and cyanobacteria: mg/l).**

## 4. Discussion

### 4.1. Key drivers of interannual variability in lake water quality

In the study lake, key water quality predictors were values observed the previous summer. Indeed, for lake TP concentration, this was the only predictor variable selected (Section 3.1). The strength of this annual autocorrelation, together with relatively low interannual variability in lake water quality (Fig. 8), are likely the reasons why the seasonal naïve forecast performed only slightly worse than the GBN, and even slightly better for chl-a (Section 3.3.3).





Aside from high temporal autocorrelation, we found positive relationships between lake TP concentration and chl-a and cyanobacteria, as widely documented elsewhere (Rousso et al., 2020). We also found a decrease in cyanobacteria as lake colour increased, again a previously documented effect (Section 3.1). No link was seen between lake colour and chl-a however, perhaps due to quality issues with the colour data before 1998 (Section 2.3), whilst cyanobacteria data were only available from 1996 and so missed the colour step-change. Although we found some evidence for relationships between weather variables (wind and precipitation) and water quality, subsequent analysis suggested the relationship was not strong enough to make it worth including weather nodes in the GBN, as the improvements in predictive performance were marginal (for lake colour) or absent (Section 3.3). Results were relatively robust to the temporal aggregation window: statistical analyses using a shorter and more causally-plausible temporal aggregation resulted in very similar relationships being highlighted. The exception was that higher rainfall and discharge may result in lower cyanobacteria peaks (Section 3.1.2), probably due to flushing, a relationship which was not accounted for in the GBN using 6-monthly aggregation and a potential area for improvement.

The lack of a temperature effect on algal biomass or cyanobacteria is interesting, as we might expect warmer summers to be accompanied by more intense blooms. However, results fit with a number of studies which found that warming effects were minor compared to nutrient effects (Lürling et al., 2018; Robarts & Zohary, 1987), and that water column stability was a key driver of cyanobacteria dynamics in dimictic lakes (Taranu et al., 2012), with wind playing a more dominant role than seasonal air temperature (Huber et al., 2012; Yang et al., 2016). We did however find a strong air temperature effect on within-year variation in chl-a and to a lesser extent cyanobacteria (Section 3.1.2), likely because within-year variability is large compared to intra-annual variability and follows a systematic seasonal pattern. When looking in more detail at some of the BN studies in which relationships were identified between air temperature and algal variables (Couture et al., 2018; Moe et al., 2019; Rigosi et al., 2015; Shan et al., 2019; Williams & Cole, 2013), the observations included in the training data in these studies were not annually aggregated, and so both with- and between-year variability were included. This may be appropriate if the aim is to look at algal dynamics within a year. However, it may not be appropriate for predicting inter-annual variation or longer term prognoses, as our analyses suggest different factors may be responsible for within-year versus between-year variability. Although temperature is certainly likely to be important in many areas, it seems likely that a number of studies will have over-estimated its importance.

As with all data-driven models, the quality of our model strongly relies on the availability and quality of the data, and in this regard we see potential for a number of improvements:

- Although the lake has a long history of monitoring, the training dataset is very small for a data driven model (≤ 39 data points). The lake showed low inter-annual variability, with gradual changes over time and few extreme events. Statistical power in a multivariate analysis is therefore limited.





- Peaks in cyanobacteria were defined by a single point, as in WFD classification, using relatively low frequency monitoring. This approach is non-robust, and it would be preferable to have higher frequency sampling and to then define peaks using, for example, the mean of a number of consecutive highest points.

- We only used data from a single point in the lake, whilst lake water quality can have high spatial variability. In Vanemfjorden, for example, there were bathing bans in place from 2000-2007, and yet the cyanobacteria data from the monitoring point is not particularly high during this period. There is some limited data available from elsewhere in Vanemfjorden, which could help improve the model, as well as remote sensing products, which are increasingly being used in cyanobacteria bloom prediction (Bertani et al., 2017; Stumpf et al., 2012).

- Additional variables could have been considered in the feature generation and selection, e.g. radiation, water temperature, and water column stability indices, although at the expense of increasing model complexity.

Overall, the GBN developed produces predictions which are almost entirely reliant on conditions observed during the previous summer. Despite the short residence time of the lake, if TP concentrations are buffered by internal sediment P release, seasonal algal peaks are not temperature limited, and water column stability is relatively insensitive to seasonal wind and temperature (e.g. because the water column is regularly mixed under normal summer conditions), then this rather simple model may be appropriate. All these things are plausible in this shallow lake with a long history of eutrophication. However, it is also likely that our model was limited by the underlying data used to identify relationships and for training, as mentioned above, and, for cyanobacteria, by the 6-month temporal aggregation window used. As an example of the limitation of the model, any events which happened during the previous winter are not currently taken into account when making forecasts. However, there is a general consensus that a large flood in winter 2000 caused a large input of TP to the lake and was responsible for the cyanobacterial blooms that occurred in subsequent years (Haande et al., 2011). Our "bottom up" approach to developing the predictive model meant that, as we did not find a relationship between winter discharge and lake TP concentration, it was not included in the model. Whilst this bottom up approach ensures that the model is not affected by pre-conceived (but potentially incorrect) beliefs, it also means that rare but perhaps important relationships are not included. In this case, incorporating expert knowledge could increase the usefulness of the predictions, in particular the impacts of extreme events. An alternative and much more time-consuming approach could be to include process-based model simulations to increase the size of the training data, assuming a robust model could be set up which adequately captured interannual variability. The BN could then be used as a "meta-model", as has been done previously at the site in the context of longer-term climate and land use change studies (Couture et al., 2018; Moe et al., 2019). However, process-based lake models typically only predict chl-a, and so cyanobacteria forecasts would still rely on empirical relationships from the data or expert knowledge.



## 4.2. Continuous GBNs for water quality prediction

Once a GBN is defined, it is straightforward to produce probabilistic predictions for water quality variables of interest, given knowledge or forecasts for a number of the remaining variables (not relevant here, but these could include, for example, seasonal climate or streamflow forecasts). Predicting the probability of lying within a WFD status class is also straightforward, and it is easy to update the training dataset using new data. These features make the approach well-suited to forecasting. In terms of performance, our GBN was modest in its prediction abilities, with $R^2$ values between 0.37 (cyanobacteria) and 0.75 (colour) and classification error rates of between 13 and 32%. As discussed above, performance was likely limited by the nature of the lake and the data available for training, but we believe the approach itself was highly promising, and would likely result in a more powerful forecasting tool in lakes or rivers which showed higher inter-annual variability and sensitivity to seasonal discharge and climate, or if used for forecasting at shorter timescales (e.g. within-year).

One of the great benefits of the GBN approach over traditional discrete approaches is that we avoid discretization and associated information loss. Our GBN could be purely parameterised in a physically-plausible way using observed data, despite the small dataset. This was not the case with a discrete BN (Sections 3.2.2 and 3.3.1), likely due to the small sample size, resulting in very low data counts in some CPT rows and associated spurious relationships. Despite the fact that the discrete BN was fit on the discrete data, the classification error of the GBN was lower for TP and cyanobacteria than the discrete BN.

However, the GBN approach has limitations which may be problematic in some settings. Firstly, the normality assumption may not be appropriate. In our study, transformed cyanobacteria and colour data almost violated this assumption, and the need for a transformation of the cyanobacteria data introduced an important bias into our back-transformed predictions. Secondly, assuming linear relationships between variables may not be appropriate. Although there was no clear evidence for non-linear relationships here (Section 3.1.1), thresholds are sometimes used to define ecological pressure-response relationships (e.g. Peretyatko et al., 2010; Scheffer et al., 1993). Overall, better performance might have been achieved with a continuous network with less stringent parametric requirements. Non-parametric or semi-parametric BN development has received a considerable amount of attention in recent years (Marcot & Penman, 2019), with a number of promising developments (e.g. Boukabour & Masmoudi, 2020; Hanea et al., 2015; Masmoudi & Masmoudi, 2019) and we expect that non-parametric continuous BN algorithms will increasingly become available in commonly-used BN software in future years. However, the simplicity of the normal approximation used in GBNs means they may remain a good first choice in many applications, particularly when datasets are small.





## 5.  Conclusions

We developed a continuous Gaussian Bayesian network (GBN) to produce probabilistic forecasts for average growing season (May-October) lake water quality (TP, chl-a and colour) and maximum cyanobacteria biovolume. The aim was to
provide early warning, in spring of a given year, of the likely conditions for the coming season. This is, to our knowledge, one of the first continuous GBNs for water quality prediction, and one of few reported continuous BNs in environmental modelling more generally. Overall, we found the GBN approach to be well-suited to seasonal water quality forecasting. It is straightforward to produce probabilistic predictions, including the probability of lying within a WFD-relevant status class. By using a continuous BN we avoided the data loss associated with discretization of continuous variables, and the GBN
could be purely parameterised using observed data, despite the small dataset (n ≤ 39, depending on the target variable). This wasn't possible using a discrete BN, highlighting a particular advantage of using GBNs when sample sizes are small, which is often the case when the focus is on interannual variability. Despite the parametric constraints of GBNs, their simplicity, together with the relative accessibility of BN software which includes GBN handling, means they are a good first choice for BN development, which should perhaps be considered more widely when data are continuous and datasets for model training
are small.

Although the GBN approach itself proved to be promising, we had more mixed success with forecasting seasonal (or inter-annual) lake water quality at our study site. Although our exploratory data analysis suggested that wind and, to a lesser extent, precipitation, exerted a control on interannual variability in lake water quality, these relationships were weak, and
overall our lake showed relatively low sensitivity to seasonal climate. Instead, the dominant source of predictability was simply the lake water quality observed the previous year, together with inter-dependencies between water quality variables. Because of this strong inertia, the GBN did not perform much better than a naïve seasonal forecast (indeed, for chl-a, the naïve forecast performed better). Potential improvements, which could make the model more powerful at predicting seasonal water quality, include incorporating expert knowledge on the likely impacts of rare events, improving the quality of the
training data (e.g. spatial representation), and expanding the training set using synthetic process-based model results. We found a much stronger weather control on within-year variability in lake water quality, and we envisage a more management-relevant forecasting tool could be developed by adapting the approach to forecast water quality at sub-annual time scales, or by applying it to forecast seasonal water quality of water bodies (rivers or lakes) that show higher interannual variability and sensitivity to seasonal climate.

**Code and data availability**

Data and scripts are available at https://github.com/NIVANorge/seasonal_forecasting_watexr, within the 'Norway_Morsa' folder.



## Author contribution

LJB developed and carried out the analysis, with input on limnological process understanding from SH, JM and FC, on
Bayesian network development from JM, and with machine learning and Python/R integration support from JES. LJB
prepared the manuscript with contributions from all co-authors.

## Competing interests

The authors declare that they have no conflict of interest.

## Acknowledgements

Funding: This work was carried out as part of the WATExR project, part of ERA4CS, an ERA-NET initiated by JPI Climate.
The work was funded by the Research Council of Norway (Project 274208), with co-funding by the European Union (Grant
690462). Many thanks to the whole WATExR project team for fruitful collaboration and discussions on the topic of seasonal
forecasting and method development. Thanks also to José-Luis Guerrero for assistance in providing meteorological data.

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
