# Peer review of "Seasonal forecasting of lake water quality and algal bloom risk using a continuous Gaussian Bayesian network"

_Hydrology and Earth System Sciences, 2021_

## Author Comment (AC1)

**Response to Reviewer #1**

Many thanks to Reviewer #1 for a very useful review. We really appreciate your careful reading of the paper. You have made some very good comments, as well as pointing out some errors and areas where our descriptions were lacking. Below, we provide comments (in red text) in response to each point raised and suggestions for changes we would make during revisions.

**Overview**

This paper presents a Gaussian Bayesian Network (GBN) for seasonal lake water quality (TP, chl-a, cyanobacteria and colour) forecasting. The GBN was developed and applied to Lake Vansjø in southeast Norway. The GBN was found well-suited for seasonal water quality forecasting and could be parameterised purely using observed data, despite this dataset being small. The forecasting performance of the GBN was assessed using a cross validation scheme, and the performance was also compared to that of a discrete BN (with the same structure) and a naïve forecast model; it was found that the 3 models performed similarly largely due to low interannual variability and high temporal autocorrelation in the study lake.

**General comment**

Overall, I think this is an interesting study that is very relevant to the HESS special issue. Although the forecasting results with the GBN were considered a mixed success at the study site, I do agree with the authors that the GBN seems to be a sensible and promising approach for water quality forecasting, and I think that by sharing all the code on GitHub the authors have provided a very useful tool for others to use and adapt. I very much enjoyed reading the paper which is both well-written and well-presented, and I believe it can be accepted after some minor revisions.

Response: Thanks for the positive feedback. It is great that you think that the data and code might be a useful tool for people. The active repository is a bit of a live (and therefore somewhat messy) workplace, and I will aim to produce a cut-down working version of the code and data used in the paper and archive it in e.g. Zenodo before final publication.

Below are my comments which I hope the authors will find useful. Lykke til!

**Specific comments**

1. My main concern is related to the discrete BN and the comparison to the GBN. It is interesting that the discrete BN did a mixed job of representing the relationships, however, I don't understand why this happens and I think this could be elaborated on further. Specific comments in relation to this:

   (i) I'm not sure how the method you used to fit the CPTs works, but considering you have a small dataset and that you are using flat priors, I'm surprised that the fitted CPT in Table 6 seems to suggest that the evidence was strong (i.e., most of parent state combinations results in low-high probabilities of around 99%-1% and 95%-5% or vice versa). Intuitively, I would have thought that the probabilities would still be influenced by the flat prior given the small dataset, but the priors have been

completely "outweighed" by the data. To me this suggest that there is something odd about the discretisation of the data and/or the target node states.

Response: It's not a problem with the discretization or target node states I don't think, but rather we didn't do a good enough job of explaining how the discrete network was fitted. Giving more weight to the prior would help smooth the fitted CPTs. For the benefit of those who are new to BNs: the simplest way of estimating the values of the CPDs at each node is just by counting how often each state of the variable occurs (conditionally on the parent states, if the variable is dependent on parents). Or you can start with a prior, which is then updated using the counts from the observed data. The priors are basically pseudo state counts added to the actual counts before normalization. Here, we used the default uniform prior in BNLearn, the very simple so-called 'K2' prior, which just adds 1 to the count of every single state. The prior doesn't therefore have much weight compared to the data. However, we could instead have specified the equivalent or imaginary sample size (iss) to be >1, and therefore use a Bayesian Dirichlet equivalent uniform prior. Then the pseudo-counts are equivalent to having observed iss uniform samples of each variable. Use of a higher iss in our discrete network would have resulted in a smoother (and probably more realistic) posterior, and I suggest that we explore increasing the iss value in a revised version of the paper. We couldn't find any rules of thumb for what value of iss might be most appropriate, so it will be a case of trying different things out and seeing whether the CPTs look more realistic. As well as exploring higher iss values, we would improve our description of the discrete BN fitting procedure.

(ii) I had a brief look at what I believe is your discretised input data files on Github (.. \BayesianNetwork\Data\DataMatrices\Discretized\), and I think these look a bit strange (although I appreciate these may not be the final version). First of all, the 'colour_prevSummer' node seems to have been given 3 states (L, M, H) contrary to what is stated in the manuscript. It also looks like the value for 'chla' does not always match the value of 'chla_prevSummer' the previous year. The same is the case for 'colour' and 'colour_prevSummer'. I would urge the authors to double check these data files and see if this possibly explain (at least partly) the results of the discrete BBN.

Firstly, thanks for digging into the data!

- You're absolutely right, we used 3 states for colour_prevSummer, not the two that we said in the manuscript. It was the only variable we used 3 for, and the hope was that in doing so we would make the most of the extremely strong correlation between colour_prevSummer and colour. We will update the text to correct this oversight.
- You're right that chla and chla_prevSummer (and all the other current vs previous summer variables) can be different for what should be the same year, well spotted. We used WFD-relevant thresholds to discretize lake TP, chl-a and cyanobacteria for the current season. For all other features, and including lake observations from the previous summer, we used regression trees to pick the thresholds to use in discretization. We do say this briefly in Section 2.6, last paragraph, but I suggest we add extra text to emphasize that (1) classification boundaries were different for the two (previous season; current season) variables; (2) that this was done because we did not have to be constrained by the need to produce management-relevant predictions when it came to discretizing the previous summer's values, and so we opted to choose the

discretization that we hoped would give us the strongest relationship between variables (as identified using regression trees), rather than WFD-relevant boundaries; and (3) that it is very possible this wasn't the best approach, and that better results might have been obtained if we had maintained the same classes through time. This relates to the more general point that discretization is subjective and time-consuming.

(iii) Finally, I wonder if it would not have been better to use expert opinion to reflect the priors in the discrete network before training, especially as you have a small dataset? To me this would seem sensible, and you already use expert opinion to inform the structure of the network. I also wonder whether you could just have discretised your GBN after it was created (in software like Netica and Genie you can specify continuous distributions and then subsequently discretise these distribution) and how the discretised model would then perform?

- Using expert opinion to decide on the priors in the discrete network would I'm sure have given better results. However, it would not then have been a "fair" test compared with the GBN.
- Your second point is a really good one, and is something we would incorporate in a revised Discussion: rather than using a GBN, a discrete network could have been used where you first assumed and specified continuous distributions, and then discretized these. This would have resolved the small sample size issue, and I think it should give near identical results to our GBN (in the case where normal distributions were assumed). Although it is a slightly clunky solution compared to just developing a GBN, and is not something we have explored ourselves (so I'm not sure how well it would work in practice), I imagine it could be a good alternative for people who use software that does not have GBN capabilities built in yet.

2. I'm not sure I fully understand how the leave-one-out cross validation works and I think it would be great if the authors could make this a bit clearer in section 2.7.1. Do you leave one data point (i.e., a year?) out at the time and then fit the GBN to the remaining data and see how well the GBN predicts the target node time-series? Or how well the GBN predicts the data point that was left out? Or something else? I also don't really understand why the cross validation is stochastic and why it was run a default 20 times.

Response: We will certainly clarify this section, and in fact it is slightly outdated compared to the final method used, which we apologize for. We in fact used k-fold cross validation, but with a high value of k (20) so that it approached leave-one-out cross validation for cyanobacteria (n=23). In short, the cross validation was repeated for each node that we wanted to estimate predictive error for (chla, cyano, TP, colour). For each of these "target" nodes in turn, the algorithm randomly assigns the time series data (i.e data for all nodes for a given year) into k subsets (20 in our case, so for the cyanobacteria data many of these subsets will just have one year in them). Then one subset is left out at a time, the BN is fitted using all the remaining subsets, and then the fitted BN is used to predict what the target node would be for the left out subset (using only the nodes that we would have data available for at the time of issuing a seasonal forecast). As the data are randomly assigned to the 20 subsets, results differ between runs (less so for cyanobacteria than the other variables), and so we repeated the procedure 20 times. Each loop through all k subsets produces a single time series of predictions, and then there are 20 sets of

these due to repeating the procedure. Each of these was then compared to observations to generate model performance statistics, and we took the mean of the model performance statistics. I hope that makes more sense?

**Minor comments**

1. Author name: I believe it should be James E. Sample. Alternatively, change JES to JS in author contributions (L670). Yes, thanks

2. L21: change "wasn't" to "was not" Ok

3. L63-64: maybe worth explaining what polymictic and dimictic lakes are; at least I'm not familiar with these terms. Will do

4. Figure 1: where is the outlet from Vanemfjorden? At Moss River? Yes (will change Mosselva to Moss River in the text, and mention this in the fig. caption).

5. L127+: Can you explain briefly why Vanemfjorden with its short residence time is more susceptible to eutrophication and cyano blooms than Storefjorden, and why it does not seem to be related to the major input source from River Hobol?

   Yes, we can add this to the text. In brief, shallow lakes tend to have stronger interaction between the water column and the (P-rich) sediments, and so P concentrations are higher in the water. In addition, the local catchment surrounding Vanemfjorden is more agricultural than the larger Storefjorden catchment.

6. L176: Should it be 1998-2013? At least in L179 you seem to suggest NIVA for 2013 as well. No, NIVA data was ok after 2007, but lower frequency (and not in winter) compared to MOVAR data. We therefore used MOVAR data rather than NIVA data where possible (can add a sentence on this).

7. L188: specify that it is River Hobol. Yes, thanks.

8. L192: Change "As the aim" to "The aim". Alternatively combine the two sentences in L192-195 and remove "therefore" on L194. Ok

9. Figure 2: You could consider plotting error bars to give an idea of the variation in the different parameters.

   We will look into this, at least for variables where growing season means are plotted (e.g. error bars of one standard deviation). For variables which are growing season maxima we could perhaps plot a single lower error bar down to the mean of the 3 or 5 highest values. I'm not sure what a meaningful estimate of variation would be for parameters which are summed over the growing season per year, so probably would not add anything for these.

10. L227-229: I'm not sure I understand why these features would have to be included as latent variables. Because they are not measured? From Figure 1, it looks like there are monitoring stations in the eastern lake basin (the same as Storefjorden?), so would you not have water quality data from here?

    This data could be included in the forecasting model if they were measurements from before the time the forecast was issued in spring (e.g. from winter). But we couldn't use Storefjorden water chemistry to forecast Vanemfjorden water chemistry in summer 2022 (for example), as we wouldn't have that data available

yet and would then also need to forecast Storefjorden water quality. Hence the need for latent variables – probably important, but have to be predicted from nodes that you would have data for *at the time the forecast was issued*. We can emphasize this a little more in the text, as a forecasting model is somewhat different to a model that is simply designed to explain/expose interrelationships in the data.

11. Table 1 and Table 2: I find it slightly confusing what features are included. Are all the features for the 6-month growing season as well as for the previous winter season (Nov-Apr), i.e., the number of features used for all variables are at least 2x13? Looking at Table 2, and if I understand the caption correctly, it looks like cyano has 8 additional features, so 34 in total (not 33).

    Yes, agree it is a little confusing. We will improve this part – perhaps easiest to just write all the features out in full.

12. Table 2: Are the features chl-a_prev, cyano chl-a and cyano_prevSummer for the lake? Yes (see the 'Description' column)

13. L293-300: I think this would be better presented as a table, where you clearly state what is defined as Low and High in the model. The specific comments related to the water quality parameter in question could then be added in a separate column (e.g., that L and H for TP is in fact lower and upper moderate and so on). Good idea.

14. L304+: I don't follow this part of the discretisation process and why you get unbalanced class sizes. Are the variables still transformed in the discrete version and fairly normal?

    We will expand on this section. The regression trees, regressing a parent (independent variable) against a child (dependent variable) node, were used to pick out class boundary divisions for the parents. However, sometimes the regression tree division resulted in a split of the data where most were in one class, and just a couple of points were in the other class. No, the variables weren't transformed in the discrete network (that being one of the benefits of discrete vs GBN).

15. L348+ and Figure 3: Is the relationship between number of calm days and TP negative? To me it looks like the two are positively correlated.

    Thanks, typo. There was a negative correlation with wind speed, not number of calm days.

16. L355: Are wind speed (winter_wind) and TP(PS) positively correlated?

    Yes, but given there is no way that lake TP concentration in e.g. summer 2020 can affect wind in winter 2020/21, we suspect this is correlation but not causation. I hope that came out ok in the text?

17. Figure 3-6: What are the bell-shaped curves and how were they derived?

    Thanks, will add description to fig. caption (they are probability density functions approximated using kernel density estimation).

18. Figure 7: Is TP_prev supposed to be linked to chl-a_prev? If so, should chl-a_prev not have a beta1_TP_prev coefficient? Very well spotted, thanks!

19. L456: Should it not say: "For parentless nodes…"? Some of your nodes are both parent and child nodes (e.g. lake TP is the parent of lake chl-a but the child of TP_prev). Yes, thanks.

20. L526: As you say, this bias in cyano is likely due to the box-cox transformation. Rather than the mean, would it not have been better to use the median (or mode)? Also, did you calculate the mean before or after back-transformation?

    The GBN prediction (mean of the cyanobacteria CPD on the Box-Cox transformed scale) was back-transformed to produce the forecasts on the original cyanobacteria data scale. However, extra reading (https://otexts.com/fpp2/transformations.html) has made us realise that the back-transformed numbers are in fact medians on the original data scale, not necessarily means, and in cases where the median and mean are different, a straight back-transformation introduces bias. We will explore using a bias-adjusted back transformation (using the formula given in the above link) to calculate forecasted mean cyanobacteria instead. This should reduce the bias, and it will be interesting to see how much by and whether it changes any of the conclusions.

21. L656: change wasn't to was not. Ok

---

## Author Comment (AC2)

**Response to Reviewer #2**

Many thanks to Reviewer #2 for a very useful review. Below, we provide comments (in red text) in response to each point raised and suggestions for changes we would make during revisions.

**General comments:**

This study presents an application of a Continuous Bayesian Network (CBN) to seasonal (6-month average) algal forecasting in a northern lake. This is likely the first use of CBN for this purpose. In general, the model performs similarly to a traditional (discretized) BN and a naïve model (using the mean from the previous year). It could be a good fit for this special issue, but I do have several concerns, as outlined below.

I'm not really sure that there is a strong contribution, as the CBN does not perform particularly well.

Response: Studies using Bayesian Networks (BN) are dominated by discrete networks (as we lay out in the introduction). However, I think one of the main points of the paper is that we found that a very simple continuous BN, a Gaussian BN (GBN), had many advantages compared to a discrete BN: it was much quicker and less subjective to develop, and the fitted conditional probability densities (CPDs) at each node were more robust. I hope that just demonstrating this simple, easy-to-use alternative to discrete BNs for water quality modelling is an important contribution to the field, given the booming popularity of BNs in environmental modelling. If you agree, then we can try to highlight this point more throughout the text (abstract, introduction, discussion, conclusions). You are of course right that a challenge in arguing the case is that the GBN performance wasn't very impressive in this case study. But it performed as well as/better than the discrete BN, and I do think that other groups looking to develop BNs may be interested in our experiences. Particularly if we tidy up our scripts (which we intend to do), and make them available via e.g. Zenodo.

Also, the model appears to be based on existing software (an R package), so there isn't new methods development. If the objective of the study is to provide a thorough demonstration of CBNs for algal bloom modeling, that could potentially be an important contribution. In this case, I'd like to see more demonstrations of how the CBN approach (e.g., Figure 7) can be advantageous for studying a system or supporting management. In my opinion, the current discussion is too focused on skill assessment (e.g., R2), which probably doesn't do justice to the CBN approach. Also, probabilistic predictions using various linear covariates can also be obtained through multiple linear regression (frequentist or Bayesian), so why use a CBN? I think there are potentially good reasons for using a CBN, but they aren't compellingly demonstrated in the current manuscript.

Response: We think that the focus on seasonal forecasting of lake water quality is also an important objective. Seasonal forecasting of water quality is in its infancy, despite its potential management relevance. We could only find 3 existing papers which looked at seasonal water quality forecasting – two were focused on river water quality and the third was part of the project which lead to this submission (and thanks for pointing out the Lake Erie seasonal forecasting, we hadn't been aware of that; do you know if there is an associated paper?).

And then you're right, that another objective is to demonstrate the use of a GBN for seasonal forecasting. I do think a thorough skill assessment is important when you are trying to develop a forecasting tool. However, you are right that the BN approach has many other benefits, and we could easily add something to demonstrate some of these. The GBN we developed was used in a prototype seasonal forecasting tool for this lake, and perhaps adding a new section (e.g. at the end of Section 3) which focused on the use of the GBN for management would help do more justice to the value of the BN approach? E.g. we could add: (1) a figure with an

example forecast for a historic year for the lake, to show the kinds of probabilistic information included (which would be harder to obtain from standard multiple linear regression methods), as well as the text interpretations we developed with local managers; (2) a brief discussion of how this kind of forecast could support management. Otherwise, it's worth noting that the CBN approach shares the same (well-documented) benefits as the discrete BN approach, which are already mentioned in the introduction (line 76 - 83).

Also, I'd like to see more discussion of how this effort compares to other CBN (or BN) applications for water quality or environmental sciences, more broadly.
Response: We can certainly add to the literature review in the introduction/discussion sections.

**Major comments:**

The paper includes a tangential analysis on making predictions at smaller time scales (e.g., Lines 208-215). I recommend removing this material, as it doesn't seem relevant to the main focus of this paper (no CBN was used). Furthermore, this additional analysis doesn't provide new insights (that aren't available through existing phytoplankton literature). It seems a bit "tacked on". If you do keep this analysis, the data should be presented (as in Figure 2 for the six-month model).
Response: We suggest moving these additional analyses (and section 3.1.2) to an appendix, as well as adding figures to present the data. I think it is nice information to include in the paper, so would rather not remove it altogether: the "pre-peak" temporal aggregation backs up the choice of variables used in the (coarse) 6 month temporal aggregation, whilst the monthly analysis highlights that it is wrong to assume that those variables which are responsible for within-year variation can be used to predict between-year variation. But you are right that it is a bit of a tangent, and so moving it to an appendix should improve the paper.

The variable selection process seems ad hoc (Section 3.1.1), making it somewhat hard to follow and likely difficult to reproduce
Response: I think our variable selection process was more robust than is usual in BNs. Many BNs are developed using a top-down approach, where researchers decide in advance which variables and relationships they think are important. We think our approach achieved a good balance of combining robust statistical methods for selecting variables (feature importance analysis) with process-based knowledge, to filter out relationships which are likely spurious, or on one occasion to add in variables which we think are of particular interest. However, I think we can simplify and cut down the text in this results Section, and improve Table 4 (results of the feature importance analysis).

Some of the explanations seem questionable. For example, the article cites previous literature showing that "windier summers" are relevant, but the CBN uses winds from the previous 6 months (prior to summer), right?
Response: There has been some misunderstanding about which periods the different variables relate to (e.g. the "wind_speed" feature is mean wind speed from the May-Oct growing season, i.e. the same time period as lake water quality is summarised over). We can amend the text clarify this, and the different time periods that were used, in Tables 1-2.

I have two general suggestions. First use clear and consistent terminology that clarifies which time periods you are talking about (also use consistent notation across the different figures and tables). Second, drop wind from the 6-month analysis altogether. Much of the text is a somewhat tortuous explanation (at least for this reader) of reasons to include/exclude wind speed, while in reality, the authors readily acknowledge that wind speed is only relevant at smaller time scales (e.g., Lines 443-445: "wind would likely only have an immediate and relatively short-lived effect…"), not ~6 months in advance.

Response: It is important that the time periods referred to are clear, so we will certainly take a careful look through the text, tables and figures and amend things, as well as improving the consistency in our terminology.

Your second suggestion is perhaps not so relevant, now that we have clarified that "wind_speed" relates to the same time period as lake chemistry? I certainly would like to keep wind speed in the analysis. We know that wind is potentially important in these systems for a variety of reasons, and I think it is feasible that a generally windier May-Oct period may result in a generally less stratified lake, and therefore e.g. lower maximum cyanobacteria during that same period. However, we will certainly smooth out the text and simplify things so that it reads more easily.

**Detail-oriented comments:**

Line 11: Clarify in the abstract that you are predicting a May-October average (rather than daily predictions).
Will do

Line 20: The term "purely parameterized" is used multiple times throughout this manuscript, but I don't understand what it means or how it is justified. As noted above, the parameterization process seems somewhat ad hoc to me.
It means that the conditional probability densities in the GBN and the conditional probability tables in the discrete BN were fitted just using the data, rather than using expert knowledge. We will clarify in the text.

Line 23: Suggest clarifying what is meant by a "naïve forecast" here.
Will do

Line 44: Models for Lake Erie cyanobacteria blooms (including Bayesian models) predict the maximum bloom size months in advance.
Thanks, we'll look into. We were relying on Rousso et al. (2020) here.

Line 56: Could you explain why "colour" is particularly relevant to water treatment or provide a reference?
Will do

Figure 1: Suggest including arrows to show dominant flow directions.
Good suggestion

Table 2: Clarify what averaging periods were used.
Good suggestion, and something Reviewer #1 struggled with too, and we are planning to re-do Tables 1 and 2.

Line 273: Clarify what normality test was used.
Will do

Figure 3, 4, 5: Clarify why only certain features are shown in each figure.
OK

Table 4: The "Feature subset" column is confusing. Use consistent terminology and explain in the caption.
Agreed, will do

Line 370-371: Revise for clarity.
OK

Line 422:  Suggest "wind-related" instead of "related" for clarity.
OK

Line 458:  The term "credible" usually refers to the uncertainty in a parameter.  It could be good to present actual parameter uncertainties (e.g., credible intervals).  Also, I don't think relationships matching the simple bivariate correlations necessarily makes them "credible" in any sense.  For example, see literature on Simpson's Paradox.
Replace "credible" with "plausible". I don't know of a good way to generate credible intervals of fitted GBN parameters, but am open to suggestions. More generally, we will re-write this part in a more nuanced way, whilst still maintaining the main message that the fitted coefficients look plausible.

Line 470:   Again, I'm not sure using simple bivariate correlations to evaluate a more sophisticated model makes sense.
Response: Again, we can make this more nuanced. But we think that the main message (that the fitted CPTs in the discrete BN were less plausible than those in the continuous BN) is backed up by our exploratory data analysis, and are just a result of the small sample size used to fit the conditional probability tables.

Table 6:  To me, making some numbers bold isn't effective for highlighting unexpected results. It really depends on which particular pair of numbers is being compared.  Also, I wouldn't describe some of these relationships as a "physical" response.
Response: Agreed, we will re-do this table, or consider replacing it with a figure showing CPTs for the whole network (the numbers will change anyway, as we plan on exploring different ways of specifying the priors in the discrete BN).

Line 569:  This statement seems too strong and/or requires clarification.
Response: The clarification came in the preceding paragraph, but do you disagree? But we can certainly qualify the statement, e.g. adding to the end of the line: "…a number of studies will have over-estimated its importance, by assuming that the within-year relationship between temperature and algal dynamics can be used to infer future algal responses to increases in summer temperature under climate change", or similar.

Line 644:  This is clearly true (based on the general nature of a GBN), but it wasn't really explored in this study.  I'm not sure why it is a conclusion.
Response: One of the main points of the paper is that continuous BNs, which have benefits over discrete BNs that go beyond pure performance, should be considered more widely when people are developing BNs. Of course you're right that we didn't focus on this aspect, so we can re-write to make it clear that it was a general motivation for the study.

Line 659:  This seems like a bit of a stretch.  I'm not sure that any "expert" can predict an extreme event ~6 months in advance.  Maybe the authors mean something else, but I can't imagine what.
Response: Where data is lacking, BNs are often developed using "expert knowledge", where researchers decide, for example, that extra nodes should be included, and on the CPT or CPDs at certain nodes (rather than using data counts or fitted distributions). This means that we could, for example, have added in a "winter discharge" – "growing season lake TP" relationship, and decided on the coefficients that define that relationship using a best guess. So the expert knowledge in this case is not about predicting an event 6 months in advance (which would indeed be a stretch), but adding in a best guess of expected (but largely unobserved) behaviour, in an attempt to make the model better able to predict outside the bounds of the training data. We can modify the text on lines 599-600 and 659 to clarify this.

---

## Author Response (AR2)

**Response to Reviewers**

**Reviewer #1**

**RC1**

**General comment**

Overall, I think this is an interesting study that is very relevant to the HESS special issue. Although the forecasting results with the GBN were considered a mixed success at the study site, I do agree with the authors that the GBN seems to be a sensible and promising approach for water quality forecasting, and I think that by sharing all the code on GitHub the authors have provided a very useful tool for others to use and adapt. I very much enjoyed reading the paper which is both well-written and well-presented, and I believe it can be accepted after some minor revisions.

Change made: a cut-down and tidied version of the code and data used in the paper has been made available at https://github.com/LeahJB/gbn-vansjo. I will archive this (Zenodo) and add a reference and doi to the paper once the revision process is finalized.

**Specific comments**

1. My main concern is related to the discrete BN and the comparison to the GBN. It is interesting that the discrete BN did a mixed job of representing the relationships, however, I don't understand why this happens and I think this could be elaborated on further. Specific comments in relation to this:

(i) I'm not sure how the method you used to fit the CPTs works, but considering you have a small dataset and that you are using flat priors, I'm surprised that the fitted CPT in Table 6 seems to suggest that the evidence was strong (i.e., most of parent state combinations results in low-high probabilities of around 99%-1% and 95%-5% or vice versa). Intuitively, I would have thought that the probabilities would still be influenced by the flat prior given the small dataset, but the priors have been completely "outweighed" by the data. To me this suggest that there is something odd about the discretisation of the data and/or the target node states.

Change made: As described in our response to reviewer #1, this wasn't a problem with the discretization of the data, but the weighting of the prior. We looked into the imaginary sample size (iss) parameter in more detail, and experimented with different values. As described in our original response, this parameter effectively controls the prior's weight compared to the data counts. In our original submission we just used bnlearn's default value of iss = 1. However, we found that using a larger value of 15 smoothed the CPTs enough that at least some of the unexpected behaviour was removed. i.e. our new network suffers much less from over-fitting due to a stronger weighting of the prior.

We have provided a better description of our method for fitting the discrete network's CPTs, including a mention of the iss parameter, in the last paragraph of Section 2.6.2. As this involved updating the fitting of the discrete BN, we have also updated

all the stats and parameters associated with the discrete network throughout the text (including the cross validation results).

(ii) I had a brief look at what I believe is your discretised input data files on Github (.. \BayesianNetwork\Data\DataMatrices\Discretized\), and I think these look a bit strange (although I appreciate these may not be the final version). First of all, the 'colour\_prevSummer' node seems to have been given 3 states (L, M, H) contrary to what is stated in the manuscript. It also looks like the value for 'chla' does not always match the value of 'chla\_prevSummer' the previous year. The same is the case for 'colour' and 'colour\_prevSummer'. I would urge the authors to double check these data files and see if this possibly explain (at least partly) the results of the discrete BBN.

Changes made: we have clarified in section 2.6.2 (first paragraph) that three classes were used for colour\_prevSummer, with justification. In this same paragraph, we have improved our description of the method for discretizing using regression trees, and added text to emphasize that chla and chla\_prevSummer (and all the other current vs previous summer variables) can be different for what should be the same year.

(iii) Finally, I wonder if it would not have been better to use expert opinion to reflect the priors in the discrete network before training, especially as you have a small dataset? To me this would seem sensible, and you already use expert opinion to inform the structure of the network. I also wonder whether you could just have discretised your GBN after it was created (in software like Netica and Genie you can specify continuous distributions and then subsequently discretise these distribution) and how the discretised model would then perform?

**Changes made:**

- As we said in our original response, we agree that using expert opinion to decide on the priors in the discrete network would likely have given better results. However, as it wouldn't have been a fair test compared to the GBN, we didn't explore this when revising.
- We have revised the Discussion (Section 4.2, end of 3rd para), to mention that, rather than using a GBN, a discrete network could have been used with specified probability distributions, which certain software then discretizes. This should give near identical results to our GBN (in the case where normal distributions were assumed), and could be a good alternative for people who use software that does not have GBN capabilities built in yet.
- 2. I'm not sure I fully understand how the leave-one-out cross validation works and I think it would be great if the authors could make this a bit clearer in section 2.7.1. Do you leave one data point (i.e., a year?) out at the time and then fit the GBN to the remaining data and see how well the GBN predicts the target node time-series? Or how well the GBN predicts the data point that was left out? Or something else? I also don't really understand why the cross validation is stochastic and why it was run a default 20 times.

**Changes made: We have updated the first paragraph of Section 2.7.1 to include a more detailed explanation of the cross validation procedure. We also changed the**

cross validation to be leave-one-out (rather than nearly leave-one-out, as used in our original submission).

**Minor comments**

- 1. Author name: I believe it should be James E. Sample. Alternatively, change JES to JS in author contributions (L670). Done, thanks
- 2. L21: change "wasn't" to "was not" Done
- 3. L63-64: maybe worth explaining what polymictic and dimictic lakes are; at least I'm not familiar with these terms. Done
- 4. Figure 1: where is the outlet from Vanemfjorden? At Moss River?

Change made: Added flow directions to Fig. 1, changed Mosselva to Moss River in the text, and mentioned where the outlet from Vanemfjorden is in the Fig. caption.

5. L127+: Can you explain briefly why Vanemfjorden with its short residence time is more susceptible to eutrophication and cyano blooms than Storefjorden, and why it does not seem to be related to the major input source from River Hobol?

Changed: Short addition made to text (penultimate para of Section 2.1).

- L176: Should it be 1998-2013? At least in L179 you seem to suggest NIVA for 2013 as well. Changed: made it clearer why we used MOVAR data for the period until 2012, despite having decent NIVA data from 2008 (second para of Section 2.3)
- 7. L188: specify that it is River Hobol. Done
- 8. L192: Change "As the aim" to "The aim". Alternatively combine the two sentences in L192-195 and remove "therefore" on L194. Done, thanks for spotting.
- 9. Figure 2: You could consider plotting error bars to give an idea of the variation in the different parameters.

Response: we looked into doing this, but the plot looked strange with error bars for some variables (those that had been averaged), and not for others (those which had been summed or were maxima). We would rather leave them out, as the focus wasn't really on this in this model.

10. L227-229: I'm not sure I understand why these features would have to be included as latent variables. Because they are not measured? From Figure 1, it looks like there are monitoring stations in the eastern lake basin (the same as Storefjorden?), so would you not have water quality data from here?

Changes made: updated first para of Section 2.4 to better explain the choice of features, given the aim of producing a model for operational seasonal water quality forecasting.

11. Table 1 and Table 2: I find it slightly confusing what features are included. Are all the features for the 6-month growing season as well as for the previous winter season (Nov-Apr), i.e., the number of features used for all variables are at least

2x13? Looking at Table 2, and if I understand the caption correctly, it looks like cyano has 8 additional features, so 34 in total (not 33).

Changes made: We have redone Table 1 to include all features explicitly and deleted Table 2 (which was redundant).

- 12. Table 2: Are the features chl-a\_prev, cyano chl-a and cyano\_prevSummer for the lake? Yes (see the 'Description' column in Table 1)
- 13. L293-300: I think this would be better presented as a table, where you clearly state what is defined as Low and High in the model. The specific comments related to the water quality parameter in question could then be added in a separate column (e.g., that L and H for TP is in fact lower and upper moderate and so on). Change made: removed this information into new Table 2
- 14. L304+: I don't follow this part of the discretisation process and why you get unbalanced class sizes. Are the variables still transformed in the discrete version and fairly normal?

Changed: Expanded Section 2.6.2 first paragraph to provide a more complete description of the discretization method used.

15. L348+ and Figure 3: Is the relationship between number of calm days and TP negative? To me it looks like the two are positively correlated.

Fixed typo, thanks.

16. L355: Are wind speed (winter\_wind) and TP(PS) positively correlated?

Yes, but unlikely to be a causative relationship.

17. Figure 3-6: What are the bell-shaped curves and how were they derived?

Changed: description added to figure captions

- 18. Figure 7: Is TP\_prev supposed to be linked to chl-a\_prev? If so, should chla\_prev not have a beta1\_TP\_prev coefficient? Added, thanks for spotting
- 19. L456: Should it not say: "For parentless nodes..."? Some of your nodes are both parent and child nodes (e.g. lake TP is the parent of lake chl-a but the child of TP\_prev). Changed.
- 20. L526: As you say, this bias in cyano is likely due to the box-cox transformation. Rather than the mean, would it not have been better to use the median (or mode)? Also, did you calculate the mean before or after back-transformation?

Changed: Originally when we back-transformed the cyanobacteria predictions, we did not adjust for bias introduced by the transformation. So the back-transformed value was the median, rather than the mean. We have now replaced this with a bias-adjusted back transformation (equation is here: <a href="https://otexts.com/fpp2/transformations.html">https://otexts.com/fpp2/transformations.html</a>). This results in much reduced bias in the cyanobacteria predictions, i.e. a much more realistic-looking forecast.

Because of this change in the way we calculate cyanobacteria, we have updated all the Tables and Figures in the text that relied on cyanobacteria forecasts. We have also updated the text (results and discussion) to reflect the improved cyanobacteria forecast performance statistics.

21. L656: change wasn't to was not. Done

**Reviewer #2**

**RC2**

**General comments:**

This study presents an application of a Continuous Bayesian Network (CBN) to seasonal (6month average) algal forecasting in a northern lake. This is likely the first use of CBN for this purpose. In general, the model performs similarly to a traditional (discretized) BN and a naïve model (using the mean from the previous year). It could be a good fit for this special issue, but I do have several concerns, as outlined below.

I'm not really sure that there is a strong contribution, as the CBN does not perform particularly well.

Also, the model appears to be based on existing software (an R package), so there isn't new methods development. If the objective of the study is to provide a thorough demonstration of CBNs for algal bloom modeling, that could potentially be an important contribution. In this case, I'd like to see more demonstrations of how the CBN approach (e.g., Figure 7) can be advantageous for studying a system or supporting management. In my opinion, the current discussion is too focused on skill assessment (e.g., R2), which probably doesn't do justice to the CBN approach. Also, probabilistic predictions using various linear covariates can also be obtained through multiple linear regression (frequentist or Bayesian), so why use a CBN? I think there are potentially good reasons for using a CBN, but they aren't compellingly demonstrated in the current manuscript.

Also, I'd like to see more discussion of how this effort compares to other CBN (or BN) applications for water quality or environmental sciences, more broadly.

**Changes made:**

- Made the aims of the paper clearer in the last para of the introduction
- Tried to make the novelty of the seasonal forecasting aspect more prominent (rephrasing parts of the introduction to describe the setting of the WATExR project, better explanation for the choice of variables to include in the analysis and how these would need to be replaced with seasonal climate forecasts in any operational model, more mention of this in the discussion)
- Added a section demonstrating the use of the GBN for supporting management (Section 3.4)
- Added a discussion comparing GBN and multiple linear regression in the context of the case study (Section 4.2, last para).

**Major comments:**

The paper includes a tangential analysis on making predictions at smaller time scales (e.g., Lines 208-215). I recommend removing this material, as it doesn't seem relevant to the main focus of this paper (no CBN was used). Furthermore, this additional analysis doesn't provide

new insights (that aren't available through existing phytoplankton literature). It seems a bit "tacked on". If you do keep this analysis, the data should be presented (as in Figure 2 for the six-month model).

Changed: moved to Appendix A

The variable selection process seems ad hoc (Section 3.1.1), making it somewhat hard to follow and likely difficult to reproduce. Some of the explanations seem questionable. For example, the article cites previous literature showing that "windier summers" are relevant, but the CBN uses winds from the previous 6 months (prior to summer), right? I have two general suggestions. First use clear and consistent terminology that clarifies which time periods you are talking about (also use consistent notation across the different figures and tables). Second, drop wind from the 6-month analysis altogether. Much of the text is a somewhat tortuous explanation (at least for this reader) of reasons to include/exclude wind speed, while in reality, the authors readily acknowledge that wind speed is only relevant at smaller time scales (e.g., Lines 443-445: "wind would likely only have an immediate and relatively short-lived effect..."), not ~6 months in advance.

Changed:

- We have re-done Table 1 and added a new column to clarify the aggregation period used for each of the variables.
- We have gone through the paper and made sure the notation is now consistent for each variable across the different tables and figures
- We have added a clearer justification at the start of Section 2.4 (Feature generation) for the choice of variables to include in the analysis
- Re-written Section 3.1 (Results of feature selection) in an attempt to make it less longwinded, and remove the emphasis on the wind discussions.

**Detail-oriented comments:**

Line 11: Clarify in the abstract that you are predicting a May-October average (rather than daily predictions).

Done

Line 20: The term "purely parameterized" is used multiple times throughout this manuscript, but I don't understand what it means or how it is justified. As noted above, the parameterization process seems somewhat ad hoc to me.

Changed: mentioned that expert knowledge is often used to parametrize CPTs when sample sizes are small (Introduction, discussion),

Line 23: Suggest clarifying what is meant by a "naïve forecast" here. Done

Line 44: Models for Lake Erie cyanobacteria blooms (including Bayesian models) predict the maximum bloom size months in advance. Altered the text

Line 56: Could you explain why "colour" is particularly relevant to water treatment or provide a reference? Reference added

Figure 1: Suggest including arrows to show dominant flow directions. Done

Table 2: Clarify what averaging periods were used.Added new column to the Table (and merged old Tables 1 and 2 into new Table 1)

Line 273: Clarify what normality test was used. Done

Figure 3, 4, 5: Clarify why only certain features are shown in each figure. Added text to figure captions

Table 4: The "Feature subset" column is confusing. Use consistent terminology and explain in the caption.

Amended this table slightly and the figure caption

Line 370-371: Revise for clarity. Done

Line 422: Suggest "wind-related" instead of "related" for clarity. Done (now in Appendix A)

Line 458: The term "credible" usually refers to the uncertainty in a parameter. It could be good to present actual parameter uncertainties (e.g., credible intervals). Also, I don't think relationships matching the simple bivariate correlations necessarily makes them "credible" in any sense. For example, see literature on Simpson's Paradox. Changed:

- Replaced "credible" with "plausible", and tried to make the sentence more nuanced: "Fitted coefficients for the Gaussian BN were all plausible, and matched the simple bivariate relationships between variables seen in the exploratory data analysis".
- Added 95% confidence intervals to a new Table B2 in Appendix B.

Line 470: Again, I'm not sure using simple bivariate correlations to evaluate a more sophisticated model makes sense.

Changed: re-written this paragraph (Section 3.2.2)

Table 6: To me, making some numbers bold isn't effective for highlighting unexpected results. It really depends on which particular pair of numbers is being compared. Also, I wouldn't describe some of these relationships as a "physical" response.

Changed: replaced Table 6 with a new Figure 8, which includes the fitted CPTs for the whole discrete network.

Line 569: This statement seems too strong and/or requires clarification.

Changed: added a qualification to the end of the line ("...a number of studies will have overestimated its importance, by assuming that the within-year relationship between temperature and algal dynamics can be used to infer future algal responses to increases in summer temperature under climate change")

Line 644: This is clearly true (based on the general nature of a GBN), but it wasn't really explored in this study. I'm not sure why it is a conclusion. Changed: deleted this sentence.

Line 659: This seems like a bit of a stretch. I'm not sure that any "expert" can predict an extreme event ~6 months in advance. Maybe the authors mean something else, but I can't imagine what.

Changed: added text to last para of conclusions to explain ourselves more clearly.

RC3

Quick question: How can this be considered a forecast model if it requires you to input wind speeds 6 months in advance? At best, we can only forecast wind speeds a week or two in advance.

**RC4**

I think the forecast development narrative is a bit muddled. To say that six-month-ahead wind speed forecasting "isn't quite there yet" is an understatement. If you are willing to consider 6-month-ahead wind forecasts, then why not also consider 6-month-ahead phosphorus forecasts? The latter is likely much more realistic.

Also, if the GBN can't provide any measure of credibility of the relationships (e.g., credible intervals for parameters), this is an important limitation that should be noted. I am not a GBN expert, so I can't provide guidance on how to do this. But it can obviously be done in most linear models (Bayesian or frequentist). Also, probabilistic predictions are easy to obtain from MLR models, so I'd be cautious about over-emphasizing this as an advantage of GBNs.

Overall, I'm not sure if I'll be able to recommend publication based on the proposed revisions. Of course, I defer to the editor.

**RC5**

Thank you for the additional notes. I think my last response was a bit hasty. At the same time, it's somewhat unclear why certain features (covariates) are based on observed data for the period of prediction (which can't be known at the time of forecast) and other features are based on forecasts of those features. However, I don't think this is a major sticking point, as the authors can further clarify these issues and their motivation in the manuscript.

I appreciate the authors exploring the credible intervals issue, and I think the proposed demonstration of an example forecast may be helpful. Given that that the GBN shares many of the same features as an MLR (linear relationships, Gaussian error distribution (usually), probabilistic predictions of continuous variables), it would be nice to clarify the potential advantages and disadvantages of the GBN approach. The authors provide some comparison with the discrete BN (Section 4.2), so perhaps something along these lines and with connections to your particular case study.

**RC7**

I agree it isn't necessary to create an MLR. For one thing, I suspect the predictive skill would be pretty similar to the GBN.

At the same time, it would be nice to elucidate how the GBN could be advantageous relative to more conventional linear statistical models for algal bloom forecasting (while also acknowledging GBN limitations). It's true that BNs have particular features documented in previous literature (e.g., Lines 76-83) but not all of these are unique to BN models, many weren't demonstrated in this case study, and some might be debatable for a GBN (given the linearity and distributional constraints). Perhaps one important distinction of the GBN is the multivariate structure illustrated in Figure

7. Perhaps the authors could explore and discuss this a bit further in the context of their case study.

Here are a couple of the relevant papers for Lake Erie. One uses an MLR and the other is similar to MLR in some ways (I think). So they might provide good context for this discussion, as well as the discussion of seasonal bloom forecasting, in general.

Obenour, D. R., Gronewold, A. D., Stow, C. A., & Scavia, D. (2014). Using a Bayesian hierarchical model to improve Lake Erie cyanobacteria bloom forecasts. Water Resources Research, 50(10), 7847-7860.

Ho, J. C., & Michalak, A. M. (2017). Phytoplankton blooms in Lake Erie impacted by both long-term and springtime phosphorus loading. Journal of Great Lakes Research, 43(3), 221-228.

Thanks for the interesting discussion. Good luck with your revisions.

**Changes made in response to discussions during RC3 – RC7 and related ACs:**

- We have re-written the text (Introduction, Section 2.4, Discussion) to provide a better background to our motivations for developing the seasonal forecasting tool, and to explain our choice of variables to include in the exploratory feature analysis.
- We have made some alterations to Figure 7 to make it clearer what the data sources are for the model development in this paper, and what they would be for an operational forecasting tool.
- We have added a new section to the discussion (Section 4.1.2) to clarify that seasonal climate data would be needed for met variables to be included in an operational model.
- Confidence intervals have been provided in new Table B2.
- We have added a new section (Section 3.4) which links to a prototype forecast developed for the case study site, with some discussion of how this might be useful to managers.
- Added a paragraph to the discussion (last para of Section 4.2) comparing the pros/cons of GBN vs MLR, in the context of our case study.
- Added the Ho & Michalak reference in relation to MLR vs GBN.

**Additional changes made (not in response to reviewer comments)**

When tidying up the code and data to put it on Zenodo I discovered two little bugs:

(1) When calculating the 6-month (growing season) averages or sums, the last day of the period (31st October) per year was not included in the aggregation when it should have been. A tiny omission, but for consistency I fixed it. As the discretization of lake colour was based on the 66th percentile of the data, the discretization threshold for colour changed a little. This fix resulted in slight changes to the seasonally-aggregated time series for all variables. It made a bit more of a difference to the seasonally-aggregated lake water quality values for years when there happened to be monitoring on the 31st of October, as water quality has fewer samples than the met data.

(2) The seasonal naïve forecast performance stats were calculated by comparing predictions to observations for the period 1981-2020 instead of 1981-2018 (as stated in the paper). This has been fixed.

As a result, the following changes have been made in the revised version of the paper:

- All the numbers reporting results of exploratory statistics, fitted BN coefficients and model performance change a little bit in the text, tables and figures. The changes are very small, and for the most part did not require any change in interpretation, with the exception of one or two small re-writes to a sentence in the results here or there. The main discussion points and conclusions are unchanged.
- When re-running the feature importance analysis I was surprised to discover that even the tiny changes in seasonally-aggregated values made the search for the best feature subset results quite different to those reported in the previous version of the paper (the 'Optimum' feature subsets in Table 4). I played with this a bit, and found it to be much less robust than I had appreciated before. I therefore decided to keep in the feature subset from the paper altogether (i.e. from the methods and results). It wasn't really mentioned much in the results anyway, but it means that Table 4 is now shorter, as I have removed the 'Optimum' feature subset.

Having made these changes, the numbers in the paper now match the numbers in the Jupyter notebooks on GitHub (and archived on Zenodo), so people can re-run the whole workflow and reproduce the results reported in the paper.

In addition, whilst going through the paper to fix these things, I did a minor round of proofreading to fix small things as I went along:

- Small changes to better fit the HESS house style and to tidy things up (e.g. rearranged the variables in Table 1 so they are alphabetical, moved one para from the discussion to the results the new penultimate para of Section 3.1 added missing info to the references).
- Added a couple of sentences to the end of the introduction to briefly mention the discussion.
- Added copyright info to the map in Fig. 1.
- Added info to the Code and Data availability section.